# CXCR5$^+$PD-1$^+$ follicular helper CD8 T cells control B cell tolerance

Yuhong Chen[1], Mei Yu[1], Yongwei Zheng[1], Guoping Fu[1], Gang Xin[1], Wen Zhu [1,2], Lan Luo[1,3], Robert Burns[1], Quan-Zhen Li[4], Alexander L. Dent[5], Nan Zhu[1], Weiguo Cui [1], Laurent Malherbe[6], Renren Wen[1] & Demin Wang [1,2,7]*

Many autoimmune diseases are characterized by the production of autoantibodies. The current view is that CD4$^+$ T follicular helper (Tfh) cells are the main subset regulating autoreactive B cells. Here we report a CXCR5$^+$PD1$^+$ Tfh subset of CD8$^+$ T cells whose development and function are negatively modulated by Stat5. These CD8$^+$ Tfh cells regulate the germinal center B cell response and control autoantibody production, as deficiency of Stat5 in CD8 T cells leads to an increase of CD8$^+$ Tfh cells, resulting in the breakdown of B cell tolerance and concomitant autoantibody production. CD8$^+$ Tfh cells share similar gene signatures with CD4$^+$ Tfh, and require CD40L/CD40 and TCR/MHCI interactions to deliver help to B cells. Our study thus highlights the diversity of follicular T cell subsets that contribute to the breakdown of B-cell tolerance.

[1] Blood Research Institute, Versiti, Milwaukee, WI, USA. [2] Department of Microbiology and Immunology, Medical College of Wisconsin, Milwaukee, WI, USA. [3] Chinese PLA General Hospital, Beijing, China. [4] Department of Immunology and Internal Medicine, University of Texas Southwestern Medical Center, Dallas, TX, USA. [5] Department of Microbiology and Immunology, Indiana University School of Medicine, Indianapolis, IN, USA. [6] Toxicology Division, Eli Lilly, Indianapolis, IN, USA. [7] Biomedical Research Center of South China, College of Life Sciences, Fujian Normal University, Fuzhou, Fujian, China. *email: dwang@versiti.org

During B-cell development in the bone marrow (BM), immature B cells expressing a self-reactive B cell receptor (BCR) are negatively selected by three distinct mechanisms: clonal deletion, receptor editing and anergy, to establish self-tolerance[1–6]. Clonal deletion eliminates autoreactive B cells whereas receptor editing changes their specificity. Anergy silences self-reactive B cells; however, anergic autoreactive B cells can be activated by self-antigens to produce autoantibodies. Immature B cells emerging from the BM continue to mature in the periphery, where autoreactivity of B cells is avoided through clonal deletion and anergy induction[7,8]. Mature B cells are activated upon encountering foreign antigens. With the help of T cells, the activated B cells migrate to the center of follicle to form germinal center (GC), where they undergo clonal expansion, somatic hypermutation, affinity maturation, class switching, and finally differentiate into memory B cells or long-lived antibody-producing plasma cells[9,10]. Somatic hypermutation in the GC reaction can increase the affinity of pre-existing autoreactive B cells and induce de novo autoreactivity of B cells[11,12]. Consistent with the concept that B-cell autoreactivity can be derived from GC reactions, autoantibodies in systemic auto-immune diseases are often class-switched and display high somatic hypermutation[13]. After the GC reaction, clonal deletion and receptor editing contribute to the removal of autoreactive memory B cells and plasma cells[14,15].

T follicle helper cells (Tfh cells) are a specialized subset of CD4[+] T cells that localize to B-cell follicles and provide the cognate B–T cell interaction and cytokines to facilitate the formation of GC, promote the differentiation of GC B cells into memory B or plasma cells, and drive the development of high-affinity antibodies[16–18]. Autoantibodies in autoimmune diseases often display class-switch recombination and high somatic mutation[13], indicating the involvement of CD4[+] Tfh cells in regulating autoantibody production[19,20]. In fact, excessive number of CD4[+] Tfh cells leads to spontaneous GC development and autoantibody production[20,21]. CD4[+] Tfh cells support auto-reactive B cells by recognizing either the same self-antigen as B cells or foreign antigens sharing similarities with self-antigens[22]. CD4[+] Tfh cells have been characterized by their expression of chemokine receptor 5 (CXCR5)[23], programmed cell death-1 (PD-1)[24], inducible costimulatory molecule (ICOS)[23], the cytokine IL-21[25,26], and the transcription factors c-Maf and Batf[27,28]. Their development is controlled by the master transcriptional regulator B-cell lymphoma 6 (Bcl6)[16]. Previous studies have discovered that Stat5 plays a critical role in T-cell development and negatively controls CD4[+] Tfh cell production and function in vivo[29,30]. Deletion of Stat5 in CD4[+] T cells results in an increase of CD4[+] Tfh cells and GC B cells, and leads to an impairment of B cell tolerance. STAT5 deficiency impairs Blimp-1 expression and results in elevated expression of Tfh-specific genes, such as *Bcl6*, *c-Mef*, *Batf*, and *IL-21*.

Antigen-specific CD8[+] T cells function as effector cytotoxic T cells that are normally be excluded from entry into B-cell follicles during the immune response[31–33]. However, recent studies have found that CXCR5[+] follicular cytotoxic T cells develop in response to chronic or acute viral infection[34,35]. These CXCR5[+] follicular cytotoxic T cells migrate to B-cell follicles and eradicate virus-infected CD4[+] Tfh cells and B cells. In addition, recent studies have found that CXCR5[+]CD8[+] T cells can differentiate into T helper cells that directly help B cells to form GC and produce antibodies[36,37].

In the current study, we discover that CD8[+] T cells can differentiate into a distinct CXCR5[+]PD-1[+] Tfh cell subset that shares similar gene signatures with CD4[+] Tfh cells and regulates B cell autoimmune responses and autoantibody production. Stat5 negatively regulates the development of CXCR5[+]PD-1[+]CD8[+] Tfh cells. Deficiency of Stat5 in CD8 T cells leads to an increase in CD8[+] Tfh cells, resulting in a breakdown of B-cell tolerance and autoantibody production. Our findings raise the possibility that targeting follicular CD8 helper T cells might be a new strategy to control autoimmunity.

## Results

**CD8[+] T-cell-specific deletion of Stat5 reduces CD8[+] T cells.** Previous studies have found that Stat5 negatively controls CD4[+] Tfh cell production and function in vivo[29,30]. Deletion of Stat5 in CD4[+] T cells results in an increase of Tfh cells and GC B cells, and leads to an impairment of B-cell tolerance. Here we investigated the role of Stat5 in CD8[+] T-cell development and function. We bred Stat5-deficient (Stat5[−/−]) and Stat5-floxed (Stat5[fl/fl]) mice with CD8Cre transgenic and Rosa-26-YFP (YFP) mice to generate Stat5[fl/−]CD8Cre/YFP mice. In these mice, Cre-mediated YFP expression could track the deletion of floxed Stat5, demonstrated by complete deletion of Stat5 in YFP[+] cells (Fig. 1a). Because non-CD8 T cells expressed a half-normal amount of Stat5 in Stat5[fl/−]CD8Cre/YFP relative to Stat5[+/+]CD8Cre/YFP mice, Stat5[+/−]CD8Cre/YFP mice were used as the controls. CD8Cre-mediated Stat5 deletion occurred in about 30% of CD4[−]CD8[+] (CD8 SP), but not in CD4[−]CD8[−] (DN), CD4[+]CD8[+] (DP) or CD4[+]CD8[−] (CD4 SP), thymocytes (Fig. 1b). Early development and the total number of thymocytes were largely normal in Stat5[fl/−]CD8Cre/YFP relative to Stat5[+/−]CD8Cre/YFP mice (Fig. 1c).

In the periphery, CD8Cre induced YFP expression in about 90% of CD8[+], but not CD4[+], splenic T cells, and all of YFP[+] cells were CD8[+] T cells, indicating the effective and specific deletion of Stat5 in splenic CD8[+] T cells (Fig. 1d). The percentage and absolute number of splenic CD8[+] T cells were markedly reduced compared to those of control mice (Fig. 1e, f). Due to the reduction of splenic CD8[+] T cells, the percentage of splenic CD4[+] T cells was increased in the mutant mice (Fig. 1f). As expected, the number of splenic CD4[+] T cells was comparable between Stat5[fl/−]CD8Cre/YFP and control mice (Fig. 1f). The CD4[+]Foxp3[+] Treg cells were also comparable between Stat5[fl/−]CD8Cre/YFP and control mice (Supplementary Fig. 1). Thus, CD8[+] T-cell-specific deletion of Stat5 impairs the peripheral development of CD8[+] T cells. The percentage and cell number of CD8[+] T but not CD4[+] T or B cells were reduced in Stat5[+/−]CD8Cre/YFP relative to Stat5[+/+] CD8Cre/YFP mice (Supplementary Fig. 2a). Therefore, Stat5 specifically regulates CD8[+] T-cell development in a dosage-dependent manner. Of note, the defects of CD8[+] T cells in Stat5[fl/−]CD8Cre mice were comparable to those in Stat5[fl/fl]CD8Cre mice (Supplementary Fig. 2b, c) and we utilized Stat5[fl/−]CD8Cre mice as the experimental mice.

**Lack of Stat5 in CD8[+] T cells causes autoantibody production.** We next examined the effect of CD8[+] T-cell-specific deletion of Stat5 on B-cell development and function. The percentage and number of splenic B cells were comparable between Stat5[fl/−]CD8Cre/YFP and Stat5[+/−]CD8Cre/YFP control mice (Fig. 2a). The total serum immunoglobulin (Ig) levels in 2-month-old Stat5[fl/−]CD8Cre/YFP mice were similar to those in control mice (Fig. 2b). However, the autoantigen array analysis demonstrated that compared to control mice, Stat5[fl/−]CD8Cre/YFP mice produced more autoreactive IgG and IgE, but not IgA and IgM, antibodies reactive to a broad spectrum of self-antigens, such as histones and complement complexes (Fig. 2c and Supplementary Fig. 3). Of note, the serum autoantibody levels were comparable between Stat5[+/+]CD8Cre/YFP mice and Stat5[+/−]CD8Cre/YFP mice (Supplementary Fig. 4). These results demonstrate that CD8[+] T-cell-specific deletion of Stat5 results in increased spontaneous autoantibody production.

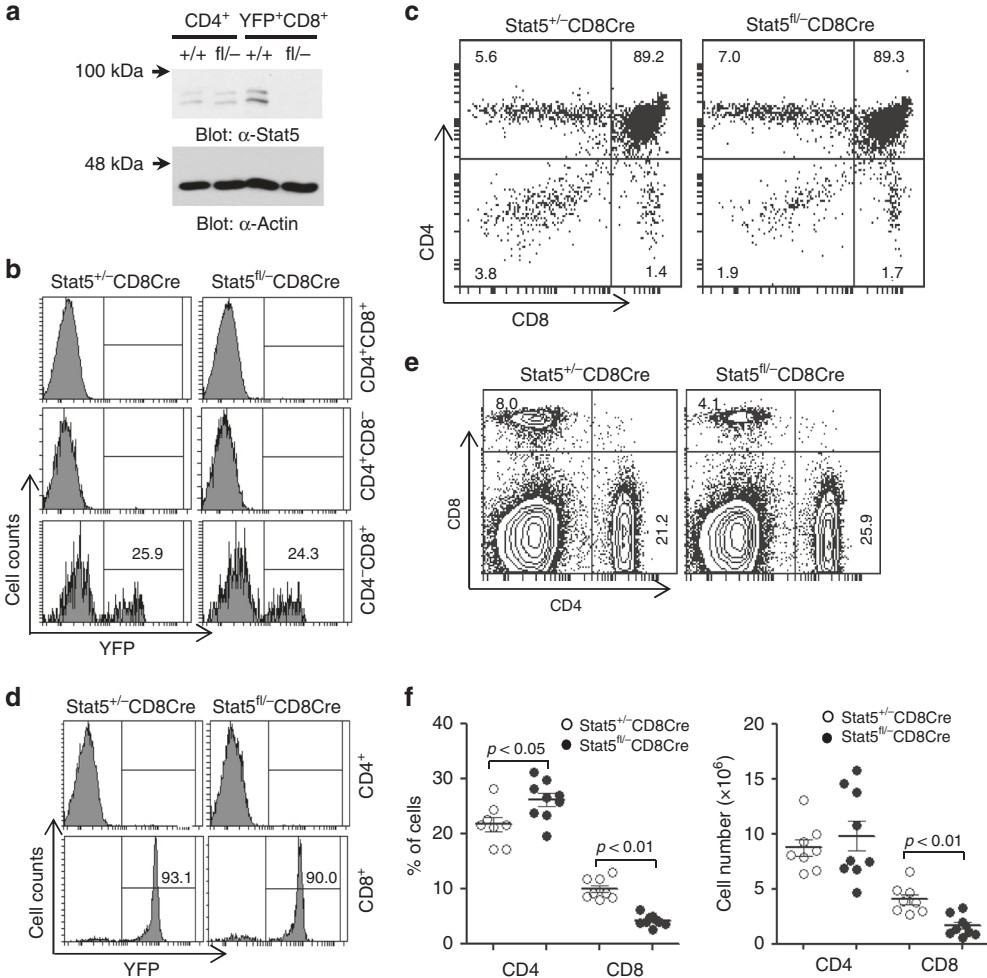

**Fig. 1** Reduction of peripheral CD8+, but not CD4+, T cells in Stat5fl/−CD8Cre mice. **a** Deletion of Stat5 in YFP+ CD8 T cells. CD4+ and YFP+CD8+ T cells from Stat5+/+CD8Cre/YFP (+/+) and Stat5fl/−CD8Cre/YFP (fl/−) mice were subjected to Western blot analysis with the indicated antibodies. **b** CD8Cre-mediated Stat5 deletion in CD4−CD8+, but not CD4+CD8+ or CD4+CD8−, thymocytes. Thymocytes from Stat5+/−CD8Cre/YFP (Stat5+/−CD8Cre) and Stat5fl/−CD8Cre/YFP (Stat5fl/−CD8Cre) mice were stained with anti-CD4 and anti-CD8 antibodies. Numbers indicate percentages of YFP+ cells in the gated CD8+ population. **c** Normal development of thymocytes in Stat5fl/−CD8Cre/YFP mice. Numbers indicate percentages of CD4−CD8−, CD4+CD8+, CD4+ CD8− or CD4−CD8+ thymocytes in the gated thymic lymphoid population. **d** CD8Cre-mediated Stat5 deletion in CD8+, but not CD4+, splenic T cells. Splenocytes were stained with anti-CD4 and anti-CD8 antibodies. Numbers indicate percentages of YFP+ cells in the gated CD8+ population. **e** Marked reduction of splenic CD8+ T cells in Stat5fl/−CD8Cre/YFP mice. Numbers indicate percentages of CD4+ and CD8+ T cells in the gated splenic lymphoid population. **f** Dot plots show percentages of CD4+ and CD8+ T cells (left) and the numbers of CD4+ and CD8+ T cells (right) in the spleen. Each dot represents an individual mouse and horizontal bars indicate mean values. Mean ± SD is shown. P-values were calculated with the unpaired two-tailed Student's t-test. Data shown are representative of 2 (**a**) independent experiments or obtained from 6 (**b**, **c**) or 8 (**d**–**f**) Stat5+/−CD8Cre/YFP and 6 (**b**, **c**) or 9 (**d**–**f**) Stat5fl/−CD8Cre/YFP mice. Source data are available as a source data file

To further study the effect of CD8+ T-cell-specific deletion of Stat5 on B cell tolerance, we crossed Stat5fl/−CD8Cre/YFP mice with IgHELsHEL transgenic mice that express a hen egg lysozyme (HEL)-specific BCR and soluble HEL (sHEL). Normally, self-antigen-induced B-cell tolerance prevents production of HEL-specific IgM in wild-type IgHELsHEL mice[38]. However, the serum level of autoreactive HEL-specific IgM was markedly increased in Stat5fl/-CD8Cre/ YFPIgHELsHEL mice relative to Stat5+/−CD8Cre/YFPIgHELsHEL control mice (Fig. 2d). Therefore, the deficiency of Stat5 in CD8+ T cells leads to increase in autoreactive antibodies in IgHELsHEL transgenic mice, resulting in the breakdown of B-cell tolerance.

**Identification of follicular CXCR5+PD-1+CD8+ T cells.** Excessive CD4+ Tfh cells leads to spontaneous GC development and autoantibody production[20,21]. It's possible that CD8+ T-cell-specific

deficiency of Stat5 affects CD8+ T cells that locate in the B cell follicles and regulate B cell function. High expression levels of CXCR5 and PD-1 are the common characteristic markers of the follicular helper cells, including CD4+ T and NKT cells[23,24,39]. Thus, we looked for the presence of follicular CXCR5+PD-1+CD8+ T cells in the spleens of Stat5fl/−CD8Cre/YFPIgHELsHEL mice. Indeed, a population of CXCR5+PD-1+CD8+ T cells was easily detected in Stat5fl/−CD8Cre/YFPIgHELsHEL relative to Stat5+/+CD8Cre/YFPIgHELsHEL, mice (Fig. 3a). The percentages of CXCR5+PD-1+ CD8+ T cells was markedly increased in Stat5fl/−CD8Cre/YFPIgH-ELsHEL relative to control mice (Fig. 3a). However, the absolute numbers of CXCR5+PD-1+CD8+ T cells were comparable in both types of mice due to the reduction of total CD8+ T cells in Stat5fl/−CD8Cre/YFPIgHELsHEL mice (Fig. 3a). Thus, the deficiency of Stat5 in CD8+ T cells results in an increased proportion of CXCR5+PD-1+CD8+ T cells in IgHELsHEL transgenic mice.

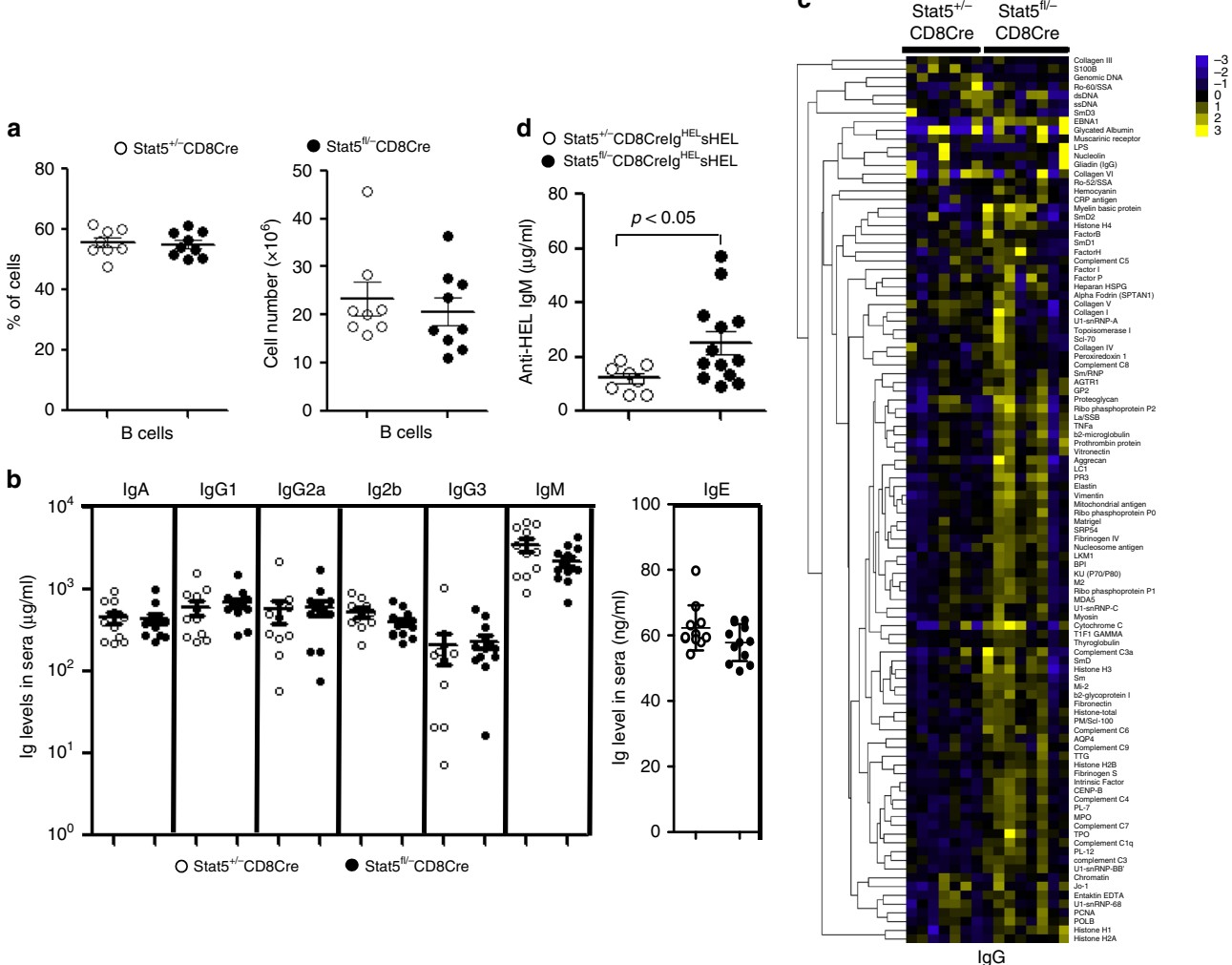

**Fig. 2** Increased autoantibody production in Stat5[fl/−]CD8Cre mice. **a** Normal numbers of splenic B cells in Stat5[fl/−]CD8Cre mice. Splenocytes from Stat5[+/−]CD8Cre and Stat5[fl/−]CD8Cre mice were stained with anti-B220. Dot plots show percentages of B220[+] B cells (left) and numbers of total splenocytes and B220[+] B cells (right) in the spleen. Each dot represents an individual mouse and horizontal bars indicate mean values. Mean ± SD is shown. **b** Normal serum Ig levels in Stat5[fl/−]CD8Cre mice. The serum levels of the Ig isotypes in 2-month-old Stat5[+/−]CD8Cre and Stat5[fl/−]CD8Cre mice were determined. Each dot represents an individual mouse and horizontal bars indicate mean values. Mean ± SD is shown. **c** Increased IgG autoantibody production in Stat5[fl/−]CD8Cre mice. The sera from 2-month-old mice were screened by the autoantigen array. **d** Self-tolerance breakdown in Stat5[fl/−]CD8CreIg[HEL]sHEL mice. Dot plots show the serum levels of HEL-specific IgM in 2-month-old Stat5[+/−]CD8CreIg[HEL]sHEL and Stat5[fl/−]CD8CreIg[HEL]sHEL mice. Each dot represents an individual mouse and horizontal bars indicate mean values. Mean ± SD is shown. P-values were calculated with the unpaired two-tailed Student's t-test. Data shown are obtained from 8 (**a**), 11 (**b**) or 7 (**c**) Stat5[+/−]CD8Cre/YFP, 9 (**a**), 13 (**b**) or 8 (**c**) Stat5[fl/−]CD8Cre/ YFP mice, 8 (**d**) Stat5[+/−]CD8CreIg[HEL]sHEL and 13 (**d**) Stat5[fl/−]CD8CreIg[HEL]sHEL mice. Source data are available as a source data file

Acute lymphocytic choriomeningitis virus (LCMV) infection causes a rapid CD8[+] T-cell expansion as well as autoimmune response[11,12]. We used the acute LCMV infection model to further study the role of Stat5 in controlling B-cell response and the generation of CXCR5[+]PD-1[+] CD8[+] T cells. Stat5[fl/−]CD8Cre/YFP or Stat5[+/−]CD8Cre/YFP mice were infected with LCMV (Armstrong). Eight days after acute viral infection, the virus in the sera, spleens, livers and kidneys were cleared in both Stat5[fl/−]CD8Cre/YFP and control mice. The frequency of antigen-specific CD8[+] T cells was comparable between Stat5[fl/−]CD8Cre/YFP and control mice, indicating that Stat5-deficient CD8[+] T cells respond normally to LCMV infection (Supplementary Fig. 5). However, more GC B cells were generated in Stat5[fl/−]CD8Cre/YFP relative to Stat5[+/−]CD8Cre/YFP (Fig. 3b). In addition, the percentage and number of CXCR5[+]PD-1[+]CD8[+] T cells were markedly more in Stat5[fl/−]CD8Cre/YFP relative to

Stat5[+/−]CD8Cre/YFP mice (Fig. 3c). In contrast, the populations of CXCR5[+]PD-1[+]CD4[+] Tfh cells were comparable between LCMV infected Stat5[fl/−]CD8Cre/YFP and Stat5[+/−]CD8Cre/YFP mice (Fig. 3c). Moreover, the frequency of CD45.2[+]CXCR5[+]PD-1[+]CD8[+] T cells in the BM chimeric mice that were transplanted with the mixture of Stat5[fl/−]CD8Cre/YFP BM cells and wild-type CD45.1 congenic BM cells was markedly higher than that in the control chimeric mice (Supplementary Fig. 6). Of note, very few CXCR5[+]PD-1[+]CD8[+] T cells were detected in Stat5[fl/−]CD8Cre/YFP and control mice without infection (Supplementary Fig. 7a, b). Thus, Stat5 deficiency in CD8[+] T cells leads to increased GC B cells and CXCR5[+]PD-1[+]CD8[+] T cells during acute viral infection.

**CXCR5[+]PD-1[+]CD8[+] T cells can directly help B cells**. A previous study identified CD8[+] regulatory T cells that suppress

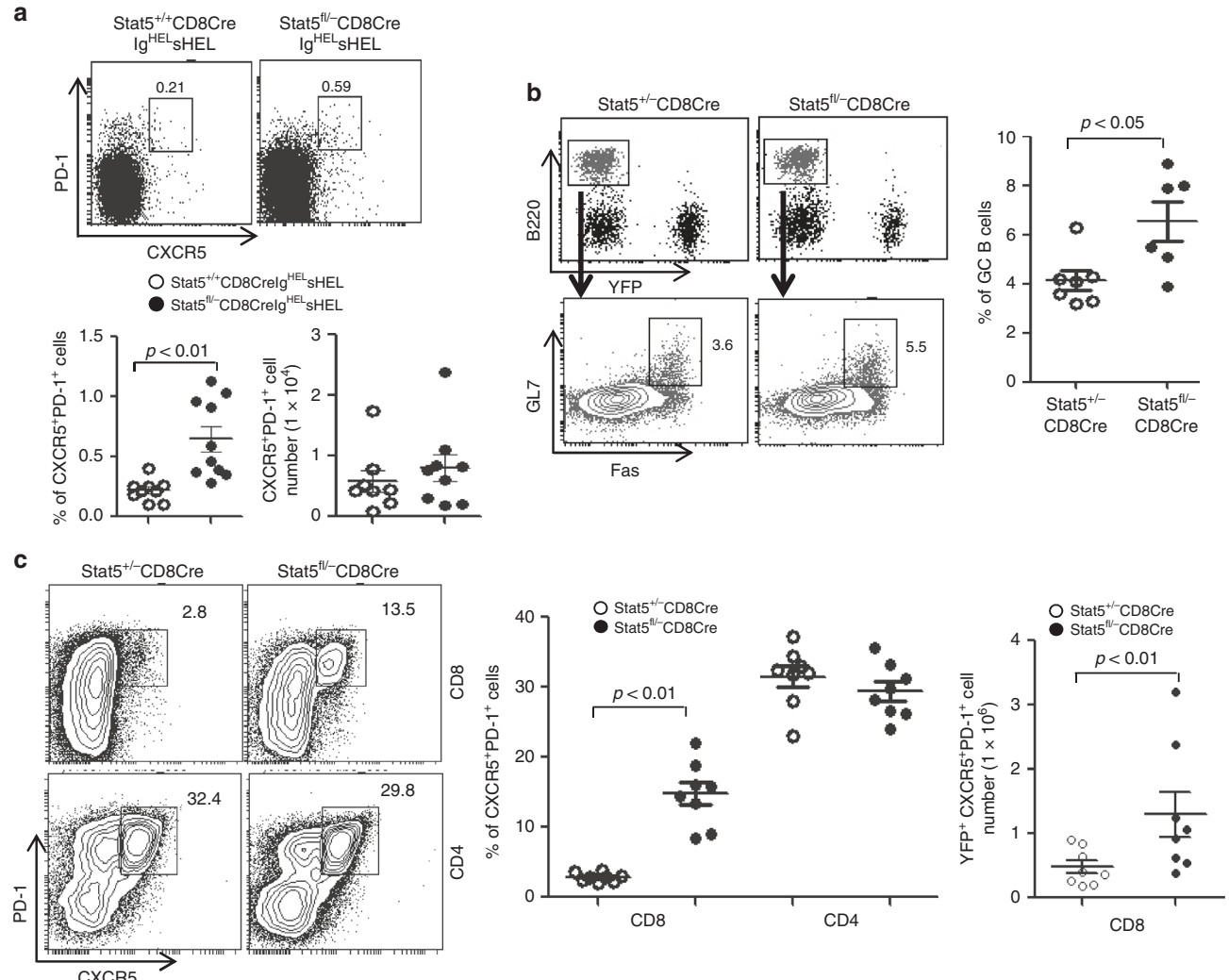

**Fig. 3** Increased CD8$^+$ Tfh and GC B cells in mice with CD8-specific deficiency of Stat5. **a** Stat5 deficiency in CD8$^+$ T cells markedly increases the percentage of CXCR5$^+$PD-1$^+$CD8$^+$ T cells in Ig$^{HEL}$sHEL transgenic mice. Splenocytes from Stat5$^{+/+}$CD8CreIg$^{HEL}$sHEL and Stat5$^{fl/−}$CD8CreIg$^{HEL}$sHEL mice were stained with anti-CD4, anti-CD8, anti-CD44, anti-CXCR5 and anti-PD-1 antibodies. Numbers indicate percentages of CXCR5$^+$PD-1$^+$ cells in the gated CD4$^+$ or CD8$^+$ population (upper and lower left) or CXCR5$^+$PD-1$^+$CD8$^+$ T cell numbers (lower right) and each dot represents an individual mouse (lower). Mean ± SD is shown. **b** Increased GC B cells in Stat5$^{fl/−}$CD8Cre mice following acute LCMV infection. Stat5$^{+/−}$CD8Cre and Stat5$^{fl/−}$CD8Cre mice were infected with LCMV. Eight days after infection, splenocytes were stained with anti-B220, anti-Fas and anti-GL7 antibodies. Numbers indicate percentages of Fas$^+$GL7$^+$ cells in the gated B220$^+$ population (left) and each dot represents an individual mouse (right). Mean ± SD is shown. **c** Markedly increased CXCR5$^+$PD-1$^+$CD8$^+$ T cells in Stat5$^{fl/−}$CD8Cre mice following acute LCMV infection. Splenocytes from the LCMV-infected mice were also stained with anti-CD4, anti-CD8, anti-CXCR5 and anti-PD-1 antibodies. Numbers indicate percentages of CXCR5$^+$PD-1$^+$ cells in the gated CD4$^+$ or CD8$^+$ population (left and middle) or CXCR5$^+$PD-1$^+$CD8$^+$ T cell numbers (right) and each dot represents an individual mouse (middle and right). Mean ± SD is shown. *P*-values were calculated with the unpaired two-tailed Student's *t*-test. Data shown are obtained from 9 (**a**) Stat5$^{+/+}$CD8Cre/YFPIg$^{HEL}$sHEL and 10 (**a**) Stat5$^{fl/−}$CD8Cre/YFPIg$^{HEL}$sHEL, 7 (**b**, **c**) Stat5$^{+/−}$CD8Cre/YFP, and 6 (**b**) or 8 (**c**) Stat5$^{fl/−}$CD8Cre/YFP mice. Source data are available as a source data file

follicular CD4$^+$ T-helper cells and thus maintain B-cell tolerance[40]. We decided to determine whether the increased B-cell autoimmune response caused by CD8$^+$ T-cell-specific deletion of Stat5 is CD4$^+$ T-cell dependent. To this end, we depleted CD4$^+$ T cells in Stat5$^{fl/−}$CD8Cre/YFP and Stat5$^{+/+}$CD8Cre/YFP mice, and then infected these mice with LCMV. The depletion of CD4$^+$ T cells completely abrogated GC B cell formation following LCMV infection in all tested Stat5$^{+/+}$CD8Cre/YFP mice (Fig. 4a). In contrast, GC B cells were generated in some of CD4$^+$ T-cell-depleted Stat5$^{fl/−}$CD8Cre/YFP mice (Fig. 4a). The production of LCMV-specific antibodies was at very low but similar levels in both CD4$^+$ T-cell-depleted Stat5$^{fl/−}$CD8Cre/YFP and Stat5$^{+/+}$CD8Cre/YFP mice (Fig. 4b), indicating that CD8$^+$ T-cell

populations are not very potent at helping antigen-specific responses. Nonetheless, the serum levels of IgG and IgE autoantibodies were dramatically higher in CD4$^+$ T-cell-depleted Stat5$^{fl/−}$CD8Cre/YFP relative to Stat5$^{+/+}$CD8Cre/YFP mice (Fig. 4c). In addition to CD4$^+$ Tfh cells, NKT cells can help the formation of GC B cells[39]. However, NKT cells were selectively deleted in both Stat5$^{fl/−}$CD8cre/YFP and control mice following acute LCMV infection (Supplementary Fig. 7c), consistent with the previous studies[41,42] and indicating that the GC formation in CD4$^+$ T-cell-depleted Stat5$^{fl/−}$CD8Cre/YFP mice was not due to the help from NKT cells. Therefore, CD8$^+$ T-cell-specific deletion of Stat5 can enhance GC formation and specifically induce autoantibody production independent of CD4$^+$ T or NKT cells.

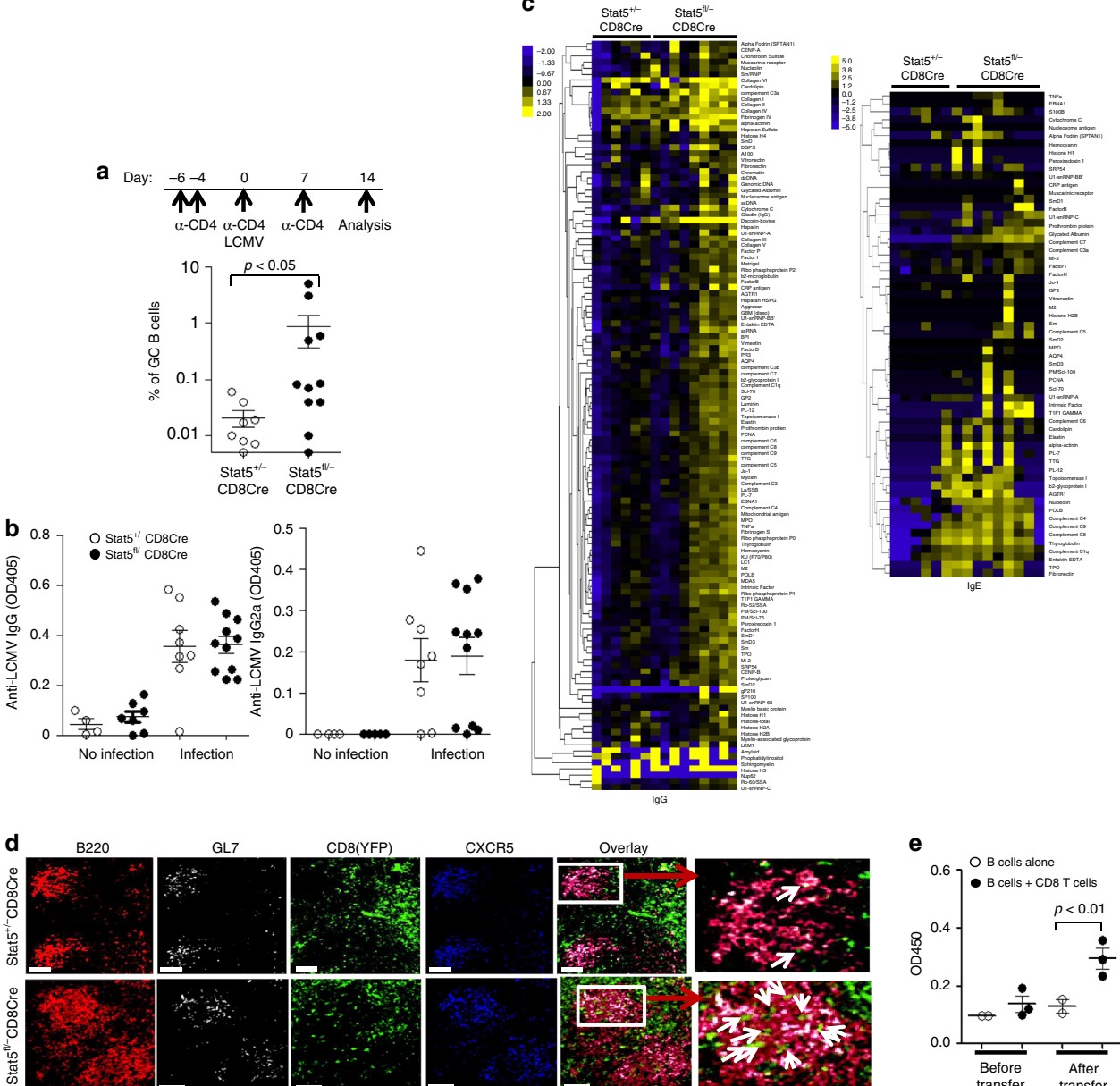

**Fig. 4** Increased autoantibody production by CD8+ T-cell-specific deletion of Stat5. **a** GC B-cell generation in CD4+ T-cell-depleted Stat5fl/−CD8Cre mice. CD4+ T-cell-depleted Stat5+/+CD8Cre and Stat5fl/−CD8Cre mice were infected with LCMV and analyzed 14 days later. The experimental scheme is shown (upper). Dot plots show percentages of Fas+GL7+ cells in the gated B220+ population. Each dot represents an individual mouse and horizontal bars indicate mean values (lower). Mean ± SD is shown. **b** The serum levels of LCMV-specific IgG and IgG2a in CD4+ T-cell-depleted Stat5+/+CD8Cre and Stat5fl/−CD8Cre mice. Each dot represents an individual mouse and horizontal bars indicate mean values. Mean ± SD is shown. **c** Markedly increased autoantibodies in CD4+ T-cell-depleted Stat5fl/−CD8Cre mice following acute LCMV infection. Following LCMV infection, the sera from CD4+ T-cell-depleted mice were screened by the autoantigen array. **d** Increased GC localization of CD8 T-cells in Stat5-deficient mice. The spleen sections from LCMV-infected Stat5+/+CD8CreYFP or Stat5fl/−CD8CreYFP mice were stained with antibodies against B220 (red), GL7 (white) and CXCR5 (blue), and imaged using the Nikon Inverted Microscope. The examples of YFP+ CD8 T cells in GC were pointed out by the white arrows. Data shown are representative of three Stat5+/+CD8CreYFP and two Stat5fl/−CD8CreYFP mice. Scale bars, 100 μm. **e** Promoting B cells to produce autoantibodies by CD8 T cells in vivo. CD8 T cells sorted from LCMV-infected wild-type mice were adoptively transferred into Rag1-deficient mice with B cells. The recipients were infected with LCMV and 2 months later, anti-nuclear antibodies (ANA) were measured. Each dot represents the level of ANA in an individual recipient before or after adoptive transfer of the indicated cells and horizontal bars indicate mean values. Mean ± SD is shown. Statistical analysis was performed with the Mann–Whitney test (**a**) or the unpaired two-tailed Student's t-test (**b**, **e**). Data shown are obtained from or representative of 8 (**a**, **b**), 6 (**c**) or 3 (**d**) Stat5+/+CD8Cre/ YFP and 11 (**a**, **b**), 9 (**c**) or 2 (**d**) Stat5fl/−CD8Cre/YFP mice. Source data are available as a source data file

Further, we determined whether Stat5-deficient CXCR5+PD-1+CD8+ T cells could directly help B cells to produce antibodies in vitro[43]. B cells and CXCR5+PD-1+CD8+, CXCR5−PD-1−CD8+, CXCR5+PD-1−CD8+ or CXCR5−PD-1+CD8+, or CXCR5+PD-1+CD4+ T cells, were sorted from LCMV-infected wild-type mice. Sorted B cells were cultured alone or with sorted CD8 or CD4 T cell subsets. Addition of sorted CXCR5+PD-1+CD8+ T cells increased the percentage of IgG1+ B cells in culture (Supplementary Fig. 8a) and the level of IgG1 in supernatant (Supplementary Fig. 8b, c) in comparison to control cultures with B cells alone, although not to the extent of CXCR5+PD-1+CD4+ T cells (Supplementary Fig. 8a–c). Of note, B cells co-cultured with wild-type or Stat5-deficient CXCR5+PD-1+CD8+ T cells produced similar levels of IgG1 (Supplementary Fig. 8c). Thus, CXCR5+PD-1+CD8+ T cells can directly help B cells to produce antibody in vitro.

Consistent with the increased CXCR5+PD-1+CD8+ T cells in Stat5fl/−CD8Cre/YFP mice during acute LCMV infection, dramatically more CD8+ T cells localized in the follicular B-cell zones, especially the GCs, of Stat5fl/−CD8Cre/YFP relative to control mice (Fig. 4d). Moreover, the adoptive transfer experiments showed that CD8 T cells promoted B cells to produce autoantibodies in the absence of CD4+ T cells (Fig. 4e). Thus, CXCR5+PD-1+CD8+ T cells might directly help B cells to produce autoantibodies in vivo.

**CXCR5+PD-1+CD8+ T cells are a novel Tfh cell sublineage.** To characterize CXCR5+PD-1+CD8+ T cells, we performed RNA-sequencing (RNA-seq) analysis to uncover the transcriptional profile of this T-cell subset. Eight days after acute LCMV infection, splenic CXCR5+PD-1+CD8+, CXCR5+PD-1-CD8+ and CXCR5−PD-1−CD8+ T cells were sorted from wild-type mice and then subjected to RNA-seq analysis. The gene expression pattern of CXCR5+PD-1+CD8+ T cells, such as the expression of inhibitory receptors, cytokines and effector molecules, memory-related receptors, transcription factors and regulators, was largely different from that of CXCR5+PD-1−CD8+ and CXCR5−PD-1−CD8+ T cells (Fig. 5a). Notably, CXCR5+PD-1+CD8+ T cells expressed significant amount of *Il21*, the critical cytokine for CD4+ Tfh cell function[25], and *Il33*, a cytokine that induces IgA and IgE production and is involved in the development of many autoimmune diseases (Fig. 5a)[44,45]. CXCR5+PD-1+CD8+ T cells expressed several other CD4+ Tfh cell signature genes, including the transcription factors *Batf*, *Maf*, *Nfatc1* and *Irf4*, the inhibitory receptors *Lag3*, *Pdcd1* and *Ctla4*, and the co-stimulatory molecule Icos (Fig. 5a). CXCR5+PD-1+CD8+ T cells dramatically reduced expression of the memory signature genes *Il7r* and *Klrg1*, and the CD4+ Tfh-cell-negative regulators, such as the transcription factors *Foxp1*, *Foxo1* and *Klf2*[46–48] and the RNA binding *Rc3h1* and *Rc3h2* that encode Roqin-1 and Roqin-2, respectively[49,50] (Fig. 5a). In contrast, CXCR5+PD-1−CD8+ T cells expressed high levels of Cytotoxic T lymphocyte (CTL) effector genes, including *Fasl*, *Gzma*, *Gzmb* and *Ifng* (Fig. 5a). Therefore, CXCR5+PD-1+CD8+ T cells are a distinct sublineage of CD8+ T cells that share similar gene signatures with CD4+ Tfh cells, and might function as helper T cells.

As expected, the Tfh-specific genes were highly expressed in CXCR5+PD-1+CD8+ T cells derived from LCMV-infected Stat5fl/−CD8Cre/YFP mice as well as non-infected Stat5fl/−CD8Cre/YFPIgHELsHEL mice (Fig. 5b, c). The expression level of *Ccr7*, the critical chemokine receptor for T-cell trafficking into splenic T-cell zones, was dramatically reduced in CXCR5+PD-1+CD8+ T cells (Fig. 5b, c). Moreover, CXCR5+PD-1+CD8+ T cells from LCMV-infected Stat5fl/−CD8Cre/YFP mice expressed even higher levels of Tfh-specific genes, such as *Il21*, *Bcl6*, *Batf*, *Btla*,

*Icos*, *Pdcd1*, *Lag3*, than corresponding wild-type CXCR5+PD-1+CD8+ T cells (Fig. 5d, e). In contrast, the expression levels of the several effector genes, such as *Fasl*, *Gzma*, *Gzmb* and *Ifng*, were further downregulated in Stat5-deficient CXCR5+PD-1+CD8+ relative to wild-type CXCR5+PD-1+CD8+ T cells (Fig. 5d, e). Comparative gene set enrichment analysis (GSEA) of the Tfh, Th1 and effector CD8 T-cell signatures showed that Tfh-specific genes were enriched in Stat5-deficient relative to wild-type CXCR5+PD-1+CD8+ T cells whereas Th1- and effector CD8 T-cell-specific genes were enriched in wild-type relative to Stat5-deficient CXCR5+PD-1+CD8+ T cells (Fig. 5f). Thus, Stat5 suppresses the expression of the Tfh-specific genes in CXCR5+PD-1+CD8+ T cells.

CD8+Tim3−Blimp1− Tfh-like cells have been identified during chronic LCMV infection, and their generation requires the transcription factor TCF1[51]. GSEA revealed that the upregulated genes in CD8+Tim3−Blimp1− relative to CD8+Tim3+Blimp1+ cells and in TCF1-overexpressing relative to control CD8+ T cells, and downregulated genes in TCF1-deficient relative wild-type CD8+ T cells were highly enriched in Stat5-deficient relative to wild-type CXCR5+PD-1+CD8+ T cells (Supplementary Fig. 9). In contrast, the downregulated genes in CD8+Tim3-Blimp1− relative to CD8+Tim3+Blimp1+ cells and in TCF1-overexpressing relative to control CD8+ T cells, and upregulated genes in TCF1-deficient relative wild-type CD8+ T cells were highly enriched in wild-type relative to Stat5-deficient CXCR5+PD-1+CD8+ T cells (Supplementary Fig. 9). Therefore, acute LCMV infection-induced CXCR5+PD-1+CD8+ T cells identified here were similar to CD8+Tim3−Blimp1− Tfh-like cells found during chronic LCMV infection, and TCF1 and Stat5 play opposite roles in controlling CD8+ Tfh cell production.

Consistent with the RNAseq results, the FACS analysis showed that several critical Tfh-related genes, including *Btla*, *Icos*, *Batf* and *Bcl6* were highly expressed in CXCR5+PD-1+CD8+ T cells, and further upregulated in Stat5-deficient cells; whereas, the expression of memory receptor *Klrg1* and inhibitory receptor *Tim3* (encoded by havcr2) was downregulated in CXCR5+PD-1+CD8+ T cells (Supplementary Figs. 10, 11). However, the protein levels of TCF1, one of the critical CD4+ Tfh-related transcriptional factors, were comparable between Stat5-deficient and control CXCR5+PD-1+CD8+ cells (Supplementary Fig. 11). The FACS analysis also confirmed that the antigen-induced IFNγ production was downregulated in Stat5-deficient relative to control CD8+ T cells (Supplementary Fig. 12). Real-time quantitative PCR (RT-qPCR) confirmed the upregulated expression of *Il-21* and *Il-33* in Stat5-deficient CXCR5+PD-1+CD8+ T cells (Supplementary Fig. 13). Thus, Stat5 negatively regulates the expression of Tfh-specific genes, but positively regulates the expression of effector T-cell-related genes, in CXCR5+PD-1+CD8+ T cells.

To further characterize CXCR5+PD-1+CD8+ T cells, we performed high-depth single-cell RNA-seq (scRNA-seq) analysis of these cells. CXCR5+PD-1+CD8+ T cells were sorted from acute LCMV-infected Stat5+/+CD8Cre/YFP or Stat5fl/−CD8Cre/YFP mice, and subjected to scRNA-seq analysis. Following quality control (QC), 500 cells from LCMV-infected Stat5+/+CD8Cre/YFP mice with median of 2160 genes per cell, and 723 cells from LCMV-infected Stat5fl/−CD8Cre/YFP mice with median of 2397 genes per cell were used for further analysis. The unbiased t-distributed stochastic neighbor embedding (t-SNE) analysis generated five cell clusters (clusters 1–5) (Fig. 6a). The clusters 1 and 2 accounted for around 80% and 15% of the total cells, respectively, whereas the clusters 3–5 accounted for very few cells (Fig. 6a–c). The quantitative transcriptional comparisons among the clusters revealed that each cluster differently expressed specific genes (Supplementary Figs. 14, 15). Of note, cluster 1

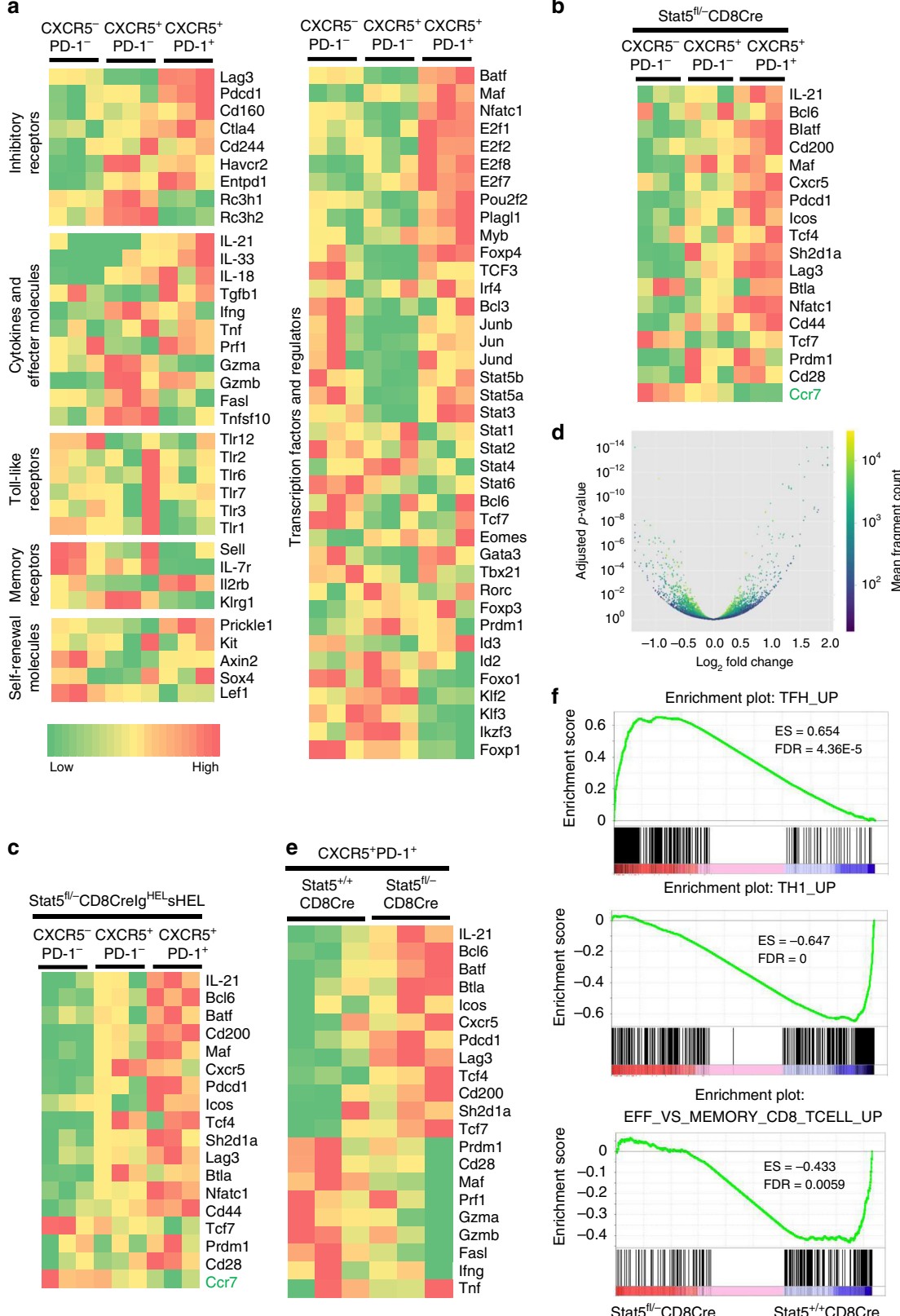

expressed high levels of the CD4+ Tfh-cell-related genes, including *Tcf7*, *Bcl6*, *Cxcr5*, *Btla*, *Il6r* and *Il7r* whereas cluster 2 expressed high levels of the CTL effector genes, such as *Gzma*, *Gzmb* and *Ifng*, chemokines, including *Ccl3*, *Ccl4* and *Ccl5*, and the CD4+ Tfh-cell-negative regulator prdm1, which is required

for effector CD8 T-cell differentiation (Fig. 6d and Supplementary Figs. 14, 15)[52,53]. The CD4+ Tfh-related genes, such as *Icos*, *Maf* and *Pdcd1*, were expressed at similar levels between the clusters 1 and 2 (Supplementary Figs. 14, 15). Among the three very minor populations, cluster 4 expressed CD8+ memory cell

**Fig. 5** Tfh-like gene expression by Stat5 deficiency in CXCR5+PD-1+CD8+ T cells. Splenic CXCR5−PD-1−CD8+, CXCR5+PD-1−CD8+, CXCR5+PD-1+ CD8+ T cells were sorted from LCMV-infected Stat5+/+CD8Cre and Stat5fl/−CD8Cre mice (**a, b, d, e**) or from Stat5+/+CD8CreIgHELsHEL and Stat5fl/− CD8CreIgHELsHEL mice (**c**), and subjected to RNA-seq analysis. **a** Differentially expressed genes in wild-type CXCR5+PD-1+CD8+ T cells relative to the indicated control subsets of wild-type CD8+ T cells. **b** Differentially expressed Tfh-specific genes in Stat5-deficient CXCR5+PD-1+CD8+ T cells relative to the indicated control subsets of Stat5-deficient CD8+ T cells. **c** Differentially expressed Tfh-specific genes in Stat5-deficient CXCR5+PD-1+ CD8+ T cells relative to the indicated control subsets of Stat5-deficient CD8+ T cells derived from IgHELsHEL transgenic mice. **d** Volcano plot of differentially expressed genes in Stat5-deficient CXCR5+PD-1+CD8+ T cells relative to wild-type CXCR5+PD-1+CD8+ T cells. **e** Differentially expressed Tfh-specific and effector-related genes in Stat5-deficient relative to wild-type CXCR5+PD-1+CD8+ T cells. Data shown are obtained from three mice of each experimental group. **f** Comparative GSEA of the Tfh, Th1 and CD8 effector cell signatures in wild-type and Stat5-deficient CXCR5+PD-1+CD8+ T cells

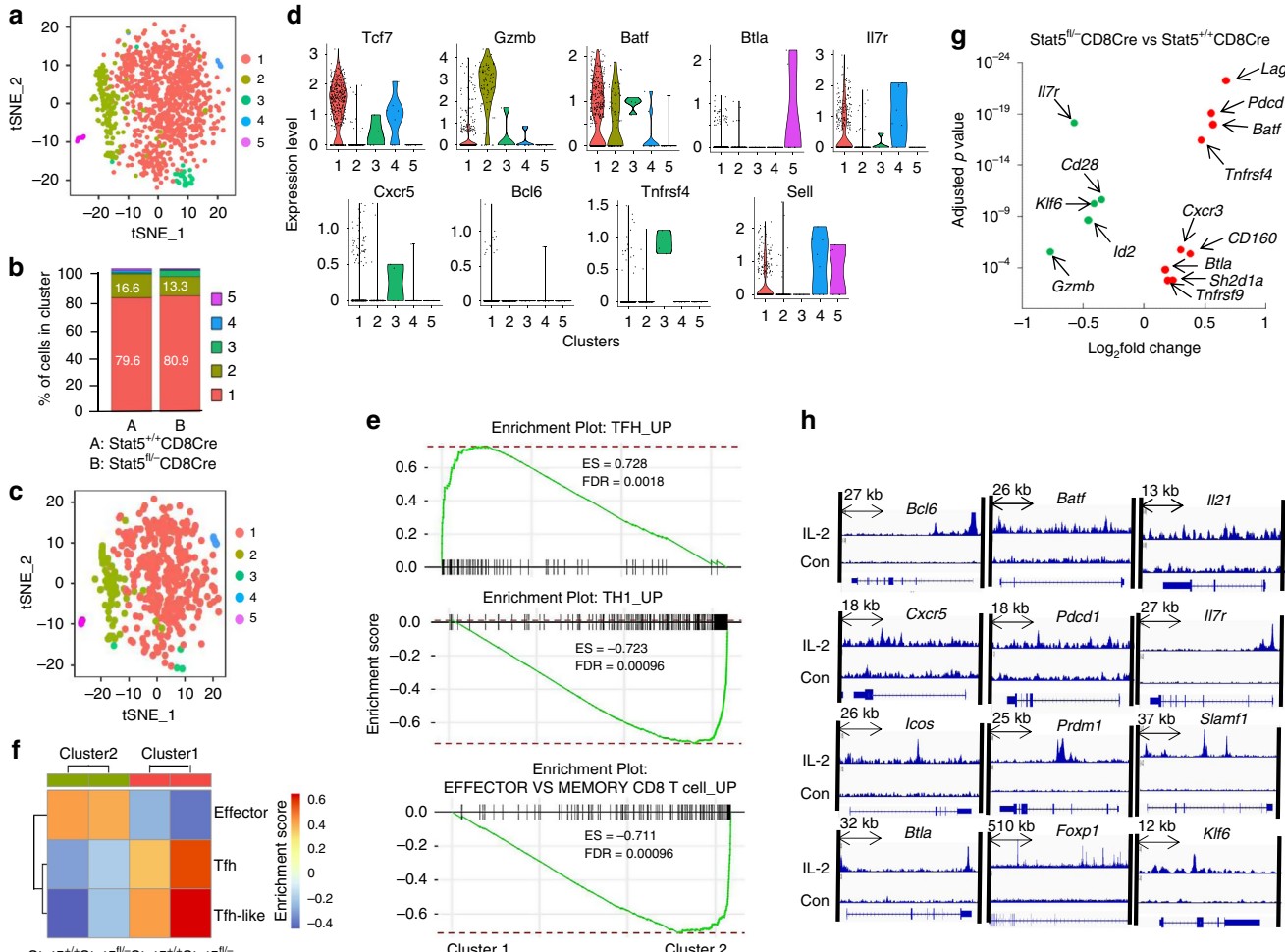

**Fig. 6** Single-cell RNA-seq analysis of CXCR5+PD-1+CD8+ T cells. CXCR5+PD-1+CD8+ T cells were sorted from LCMV-infected Stat5+/+CD8Cre/YFP or Stat5fl/−CD8Cre/YFP mice. **a** t-SNE plot of cells from Stat5+/+CD8Cre/YFP and Stat5fl/−CD8Cre/YFP mice. Each cell is grouped into one of the five clusters that are distinguished by the indicated colors. **b** Percentages of cells in each cluster within the total cells in all five clusters. **c** t-SNE plot of cells from Stat5+/+CD8Cre/ YFP mice. **d** Violin plot showing the distribution of the expression levels of Tcf7, Gzmb, Batf, Tnfrsf4, Sell, Il7r, Btla, Bcl6 and Cxcr5 in each clusters of CXCR5+PD-1+CD8+ cells from Stat5+/+CD8CreYFP mice. **e** Comparative GSEA of the Tfh, Th1 or CD8 effector cell signatures in cluster 1 and cluster 2 of wild-type CXCR5+PD-1+CD8+ T cells. **f** GSVA of the Tfh, Th1 or CD8+ effector cell signatures in cluster 1 and cluster 2 of wild-type and Stat5-deficient CXCR5+PD-1+CD8+ T cells. **g** Differential expression of Tfh-related genes in the cluster 1 of wild-type and Stat5-deficient CXCR5+PD-1+CD8+ T cells. **h** ChIP-Seq QC Alignment (Bowtie2) analysis shows IL-2-induced Stat5 DNA binding of the Tfh-related genes in CD8+ T cells. Data from GSE72565 were analyzed

signature genes, including *Sell, Il7r, Ccr7* and *Dapl1*, cluster 3 expressed *Tnfrsf4* and *Tnfrsf9*, and cluster 5 highly expressed Tyrobp (encodes DAP12), CD74 and other Class II genes (H2-Aa, H2-Ab1 and H2-Eb1, etc.), indicating these cells might be CD8+ dendritic cells (Supplementary Figs. 14–16). In addition, we performed scRNA-seq analysis of CXCR5+PD-1+CD8+ T cells from Stat5+/+ CD8Cre/YFPIgHELsHEL and Stat5fl/−

CD8Cre/YFPIgHELsHEL mice, and found that these T cells displayed similar gene expression profiles, compared to the same T cells from Stat5+/+CD8Cre/ YFP and Stat5fl/−CD8Cre/YFP mice (Fig. 6c, d and Supplementary Figs. 16–18). Thus, CXCR5+ PD-1+CD8+ T cells contain two main distinct subpopulations, the major fraction of Cxcr5hiTcf7hiBcl6hiBtlahiIcos+Maf+Pdcd1+ prdm1lo Tfh-like cells (cluster 1, ~80%) and a minor fraction of

Cxcr5$^{lo}$gzma$^+$gzmb$^+$ effector-like cells (cluster 2, ~15%) (Fig. 6d and Supplementary Figs. 14–18).

Moreover, GSEA and gene set variation analysis (GSVA) demonstrated that Tfh-specific genes were more highly expressed in the cluster 1 relative to cluster 2 of wild-type and Stat5-deficient CXCR5$^+$PD-1$^+$CD8$^+$ T cells. In contrast, the Th1- or effector CD8 T-cell-specific genes were enriched in cluster 2 relative to cluster 1 (Fig. 6e, f). Consistent with bulk RNA-seq data, the Tfh-specific genes were enriched in Stat5-deficient relative to wild-type CXCR5$^+$PD-1$^+$CD8$^+$ T cells (Supplementary Fig. 19). The expression of the Tfh-positive regulators *Batf, Pdcd1, Lag3, Tnfrsf4, Cxcr3, Tnfrsf9, Bcl6* and *Btla* was high in Stat5-deficient relative to wild-type cells (Fig. 6g and Supplementary Fig. 20). In contrast, the expression of the Tfh negative regulator *Prdm1*, the effector genes *Gzmb* and *Fasl* was low in Stat5-deficient relative to wild-type cells (Fig. 6g and Supplementary Fig. 20). The expression of Klf6, which suppresses Bcl6 expression by elevating Prdm1, was also reduced in Stat5-deficient relative to wild-type cells (Fig. 6g). Therefore, the scRNA-seq data further confirm that CXCR5$^+$PD-1$^+$CD8$^+$ T cells are largely a novel T helper cell sublineage and are negatively regulated by Stat5.

We further studied the importance of some of the genes differentially expressed in Stat5-deficient relative to control CXCR5$^+$PD-1$^+$CD8$^+$ T cells. TCF1 is reported to be essential for CD8 Tfh cell development[51]. We examined the effect of CXCR5, PD-1, Bcl6, Lag3 or ICOS deficiency on CXCR5$^+$PD-1$^+$CD8$^+$ T-cell generation. CXCR5 or PD-1 deficiency resulted in no detection of CXCR5$^+$PD-1$^+$CD8$^+$ T cells following acute LCMV infection (Supplementary Fig. 21a), demonstrating that CXCR5$^+$PD-1$^+$CD8$^+$ T cells are a real subpopulation. Further, Bcl6 or Lag3 but not ICOS deficiency impaired the generation of CXCR5$^+$PD-1$^+$CD8$^+$ T cells during acute LCMV infection (Supplementary Fig. 21a, b). Of note, all of the mice responded to acute LCMV infection to the same extend, evidenced by the observation that all mice generated comparable levels of LCMV GP33-specific CD8 T cells (Supplementary Fig. 21a). Of note, PD-1 deficiency did not affect the generation of CXCR5$^+$ICOS$^+$ CD8 T cells (Supplementary Fig. 21c), indicating that PD-1 might not be required for the generation of CD8 Tfh cells. Therefore, Bcl6 and Lag3, the genes differentially expressed in Stat5-deficient relative to control CXCR5$^+$PD-1$^+$CD8$^+$ T cells, are important for the formation of these CD8 T cells.

To demonstrate the mechanisms underlying the regulation of CD8$^+$ Tfh cells by Stat5, we analyzed IL-2-induced Stat5 DNA binding in CD8$^+$ T cells based on the published Stat5 ChIP-Seq data set GSE72565. Consistent with our findings, the ChIP-Seq QC Alignment (Bowtie2) analysis showed that most of the critical Tfh-related genes, such as *Bcl6, Batf, Il21, Cxcr5, Pdcd1, Il7r, Icos, Prdm1, Slamf1, Btla, Foxp1* and *Klf6*, were directly associated with Stat5 upon IL-2 stimulation in CD8$^+$ T cells (Fig. 6h and Supplementary Table 1). Further, we compared the Stat5 ChIP-seq data with the reported active or repressive histone marks, such as H3K4me1, H3K4me3, H3K27ac and H3K27me3, in CD8$^+$ T cells from acute LCMV-infected wild-type mice[37]. We found the Stat5 binding regions within the Tfh-related genes largely overlapped with the peaks of histone modification marks (Supplementary Fig. 22). These data suggest that Stat5 may directly bind to Tfh-related genes and negatively regulate the differentiation of CD8$^+$ Tfh cells.

**CD40L/CD40 and TCR/MHCI interaction for CD8$^+$ Tfh function.** We investigated the molecular mechanism by which CXCR5$^+$PD-1$^+$CD8$^+$ T cells regulate B-cell function. CD4$^+$ T-cell helper function depends on CD40–CD40L interaction[54,55],

and thus we studied whether the help for B cells from CXCR5$^+$ PD-1$^+$CD8$^+$ T cells depends on this interaction. We first examined the CD40L cell surface expression on CD8$^+$ T cells. After LCMV infection, CD40L was expressed markedly higher on CXCR5$^+$PD-1$^+$, relative to CXCR5$^-$PD-1$^-$, CXCR5$^-$PD-1$^+$ or CXCR5$^+$PD-1$^-$, CD8$^+$ T cells (Fig. 7a). The relatively low expression level of CD40L on CXCR5$^+$PD-1$^+$CD8$^+$ T cells was probably due to the rapid degradation of CD40L following its interaction with CD40 on antigen-presenting cells[56]. In fact, following in vitro reactivation by Phorbol 12-myristate 13-acetate (PMA) plus ionomycin stimulation, the vast majority of CXCR5$^+$ PD-1$^+$, but not CXCR5$^-$PD-1$^-$, CXCR5$^-$PD-1$^+$ or CXCR5$^+$ PD-1$^-$, CD8$^+$ T cells expressed CD40L (Fig. 7b). Further, we examined whether the disruption of CD40L/CD40 interaction would impair the ability of CXCR5$^+$PD-1$^+$CD8$^+$ T cells to help B cells. CXCR5$^+$PD-1$^+$CD8$^+$ T cells sorted from LCMV-infected mice helped wild-type, but not CD40-deficient, B cells to generate IgG1 (Fig. 7c). As expected, CD4$^+$ T cells sorted from LCMV-infected mice enhanced wild-type, but not CD40-deficient, B cells to produce IgG1 (Fig. 7c). In addition, CD40L-neutralizing antibodies prevented CXCR5$^+$PD-1$^+$CD8$^+$ or CD4$^+$ T cells from helping B cells to produce IgG1 antibodies (Fig. 7d). Thus, the ability of CXCR5$^+$PD-1$^+$CD8$^+$ T cells to help B-cell function depends on the CD40/CD40L interaction.

The CD8$^+$ T cells recognize the antigenic peptides displayed by MHC I molecules on the cell surface. CD8$^+$ T regulatory cells regulate CD4$^+$ Tfh cells through recognizing MHC Ib molecules (Qa-1) on the surface of Tfh cells[40]. We examined whether the help from CXCR5$^+$PD-1$^+$CD8$^+$ T cells for B cells required the interaction between TCR and MHC I. CXCR5$^+$PD-1$^+$CD8$^+$ T cells sorted from LCMV-infected wild-type C57BL/6J mice were co-cultured with B cells sorted from LCMV-infected wild-type C57BL/6J or single MHC I-mismatched C57BL/6J.C-H2$^{bm1}$ (bm1) mice. CXCR5$^+$PD-1$^+$CD8$^+$ T cells from C57BL/6 mice helped B cells from MHC I-matched C57BL/6J, but not MHC I-mismatched bm1, mice to produce IgG1 (Fig. 7e). In contrast, CD4$^+$ T cells from C57BL/6J mice promoted B cells from either MHC I-matched C57BL/6J or MHC I-mismatched bm1 mice to produce IgG1 (Fig. 7e).

Further, we examined whether the TCR/MHC I interaction is required for CXCR5$^+$PD-1$^+$CD8$^+$ T cells to help B cells function in vivo. CD8$^+$ T cells sorted from LCMV-infected wild-type C57BL/6J or bm1 mice were adoptively transferred into Rag1-deficient mice along with B cells from LCMV-infected bm1 mice. The recipients were infected with LCMV, and IgG production was measured 30 days later. In the recipients, CD8$^+$ T cells from bm1, but not C57BL/6J, mice promoted B cells from MHC I-matched bm1 mice to produce IgG1 (Fig. 7f). Therefore, CXCR5$^+$ PD-1$^+$CD8$^+$ T cells require the TCR/MHC I interaction to support the antibody production of B cells.

**Discussion**
CXCR5$^+$PD-1$^+$CD8$^+$ Tfh cells identified in our study are a distinct novel CD8$^+$ T cell sublineage and have an important function of controlling autoantibody production. Dysregulation of CXCR5$^+$PD-1$^+$CD8$^+$ Tfh cell development by CD8$^+$ T-cell-specific deletion of Stat5 results in the breakdown of B-cell tolerance. Of note, non-CD8 T cells express half normal amounts of Stat5 in Stat5$^{fl/-}$CD8cre experimental mice, and thus Stat5$^{+/-}$ instead of Stat5$^{+/+}$ mice are ideal controls. One important feature of CXCR5$^+$PD-1$^+$CD8$^+$ Tfh cells is the expression of CXCR5 that is normally expressed on B cells and CD4$^+$ Tfh cells and directs cells to migrate into the B-cell follicles of the spleen and lymph nodes[43,57,58]. Recently, three subsets of CXCR5-expressing CD8$^+$ T cells have been identified (Supplementary Table 2)[34,40,59]. The

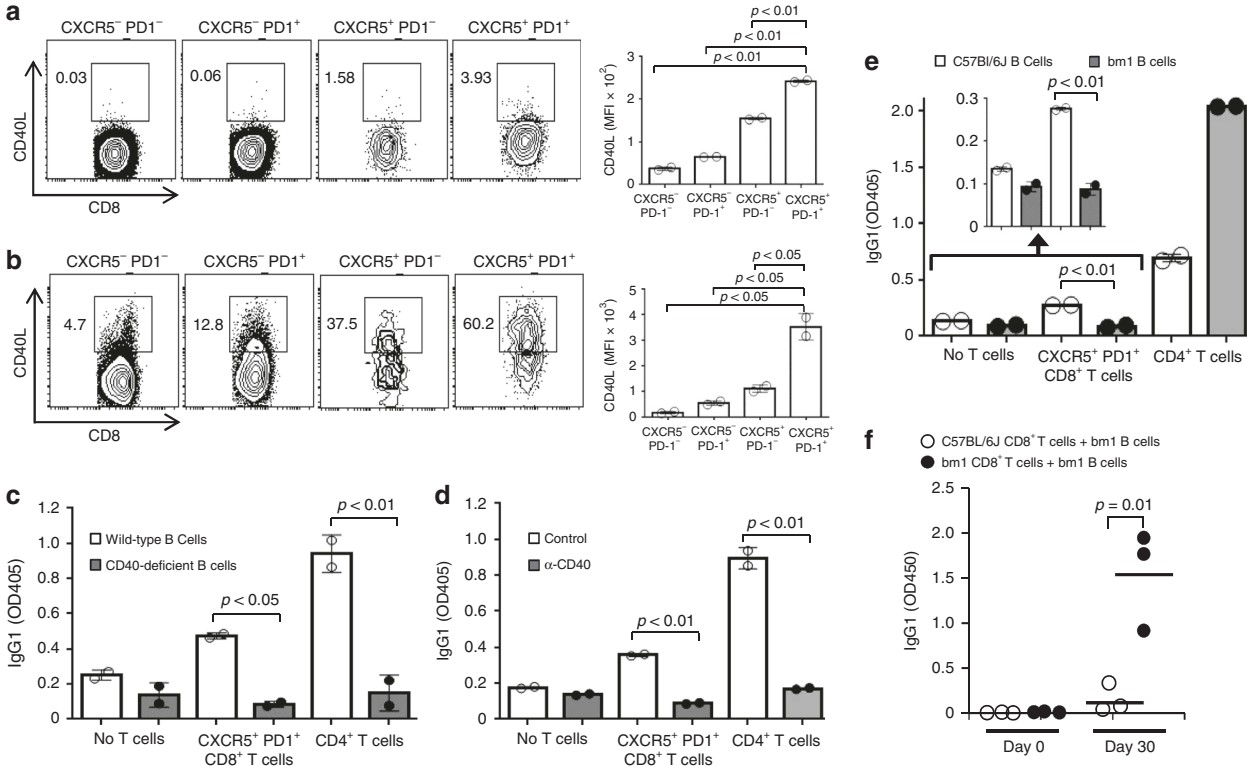

**Fig. 7** CD40L/CD40 and TCR/MHC I interactions required for CD8+ Tfh function. **a** CD40L expression in CXCR5+PD-1+CD8+ T cells. Splenocytes from LCMV-infected wild-type mice were stained with anti-CD4, anti-CD8, anti-CXCR5, anti-PD-1 and anti-CD40L antibodies. Numbers indicate percentages of CD40L+ cells (left) and bar graphs show the average MFI of CD40L expression in the indicated CD8+ populations (right). **b** CD40L expression in CXCR5+ PD-1+CD8+ T cells following activation. Splenocytes from LCMV-infected wild-type mice were stimulated with PMA/ionomycin and stained as in **a**. Numbers indicate percentages of CD40L+ cells (left) and bar graphs show the average MFI of CD40L expression in the indicated CD8+ populations (right). **c, d** Requirement of CD40/CD40L interaction for CXCR5+PD-1+CD8+ T cells to help B cells. B cells sorted from LCMV-infected wild-type or CD40-deficient mice were cultured without T cells or with CXCR5+PD-1+CD8+ or CD4+ T cells sorted from LCMV-infected wild-type mice (**c**). Wild-type B cells were cultured with CXCR5+PD-1+CD8+ or CD4+ T cells sorted from LCMV-infected wild-type mice with or without anti-CD40L neutralizing antibodies (**d**). Bar graphs show the IgG1 levels in the supernatant of the indicated co-cultures. **e, f** Requirement of TCR/MHC I interaction for CXCR5+PD-1+CD8+ T cells to help B cells. B cells sorted from LCMV-infected C57BL/6J or bm1 mice were cultured without T cells or with CXCR5+PD-1+CD8+ or CD4+ T cells sorted from LCMV-infected C57BL/6J mice and the inset shows the zoomed area (**e**). Bar graphs show the IgG1 levels in the supernatant of the indicated co-cultures. CD8+ T cells sorted from LCMV-infected wild-type C57BL/6J or bm1 mice were adoptively transferred into Rag1-deficient mice along with B cells from LCMV-infected bm1 mice (**f**). The recipients were infected with LCMV and IgG production was measured 30 days later. Dot plots show the IgG1 levels in the sera of recipients adoptively transferred with the indicated cells. Bars represent the mean values and each dot represents an individual data value (**a–e**) or recipient (**f**) and horizontal bars indicate mean values. Mean ± SD is shown. *P*-values were calculated with the unpaired two-tailed Student's *t*-test. Data shown are representative of two (**a–f**) independent experiments. Source data are available as a source data file

first subset of CXCR5-expressing CD8+ T cells are exhausted T cells and function to control viral replication in mice with chronic LCMV infection[59]. The second subset of CXCR5-expressing CD8+ T cells are cytotoxic T cells that eliminate LCMV-infected CD4+ Tfh cells and B cells in the B cell follicles[34]. It seems that these two subsets of CXCR5+CD8+ T cells function as effector CD8 T cells. The third subset of CXCR5-expressing CD8+ T cells functions as regulatory T cells that inhibit the production of autoantibodies[40]. Furthermore, CXCR5+PD-1+ CD8+ T cells expressing both CXCR5 and PD-1 have also been identified in mice with chronic LCMV infection[60]. These cells are precursors of exhausted CD8+ T cells that sustain the viral-specific CD8+ T cells during chronic infection[60]. Of note, gene expression profile analysis demonstrates that the previously identified CXCR5+PD-1+CD8+ T cells express CD4 Tfh cell-related signature genes similar to CXCR5+PD-1+CD8+ Tfh cells identified in our study[60]. However, the former CXCR5+PD-1+ CD8+ T cells express high levels of the genes related to CD8 T-cell memory precursors and hematopoietic stem-cell progenitors that are not found in our CXCR5+PD-1+CD8+ Tfh cells[60]. One more

important feature of CXCR5+PD-1+CD8+ Tfh cells identified in our study is the production of IL-21 that is not generated in the previously identified CXCR5+CD8+ or CXCR5+PD-1+CD8+ T cells[34,40,59,60]. Taken together, CXCR5+PD-1+CD8+ Tfh cells identified in our study are functionally and transcriptionally distinct from the previously identified CXCR5+CD8+ or CXCR5+ PD-1+CD8+ T cells in mice with chronic LCMV infection.

The scRNA-seq analysis showed that CXCR5+PD-1+CD8+ Tfh cells could be divided into a major subpopulation of helper-like T cells that express the signature genes of CD4+ Tfh cells, and a minor subpopulation of effector-like T cells that are enriched with effector-related genes, such as gzmb and gzma. It is likely that the minor subpopulation was a contamination of CXCR5− and CXCR5lo effector CD8+ T cells during cell sorting. When CXCR5+PD-1+CD8+ Tfh cells were markedly increased in Stat5-deficient mice, the amount of the minor subpopulation was dramatically reduced. Additional cell surface markers might be helpful of removing these cells. Nonetheless, the scRNA-seq analysis demonstrated that the CXCR5+PD-1+CD8+ Tfh cells were largely a homogenous lineage that has similar gene

expression profiles as CD4[+] Tfh cells. Our comparative GSEA and GSVA of RNA-seq data from CXCR5[+]PD-1[+]CD8[+] T cells with the gene signatures derived from the previously reported Tfh-like CD8[+] population reveals the similarities among these populations[51]. To fully understand the nature and function of CXCR5[+]PD-1[+]CD8[+] T cells, we might need to compare scRNA-Seq data of these T cells derived from different experimental situations, such as chronic LCMV and other infections.

RNA-seq analysis discovered that the expression levels of Bcl6, the master transcriptional regulator for CD4[+] Tfh cell development[61,62], and Tcf7, a critical transcription factor for early CD4[+] Tfh differentiation[63], in CXCR5[+]PD-1[+]CD8[+] Tfh cells were comparable to those in other subsets of CD8[+] T cells. However, the scRNA-seq analysis revealed that within the CXCR5[+]PD-1[+]CD8[+] T cell population, the expression of Bcl6 and Tcf7 was highly enriched in Tfh-like, but not effector-like, subpopulations. Tcf7 has been shown to play a critical role in CD8 Tfh-like cell development[51]. We found Bcl6 deficiency impaired the generation of CXCR5[+]PD-1[+]CD8[+] T cells during acute LCMV infection. Thus, Bcl6 and Tcf7 might be the potential critical transcription factors for controlling the differentiation of CXCR5[+]PD-1[+]CD8[+] Tfh cells. Importantly, several Tfh-related transcription factors, such as BATF, MAF, NFATc1 and Irf4, were upregulated in CXCR5[+]PD-1[+]CD8[+] Tfh cells. In addition, E2f1, E2f2, E2f7 and E2f8, the members of the transcription factor Foxp4/E2f family, were dramatically upregulated in CXCR5[+]PD-1[+]CD8[+] Tfh cells. The roles of these transcription factors in the differentiation of CXCR5[+]PD-1[+]CD8[+] Tfh cells warrant further studies. Further, we found Lag3 deficiency impaired CXCR5[+]PD-1[+]CD8[+] T-cell generation following acute LCMV infection, demonstrating that the genes differentially expressed in Stat5-deficient relative to control CXCR5[+]PD-1[+]CD8[+] T cells could function downstream of Stat5 to regulate the formation of these CD8 T cells. The molecular mechanism underlying the control of CXCR5[+]PD-1[+]CD8[+] T cells warrants more studies.

B-cell self-tolerance is tightly regulated and its dysregulation results in autoantibody production. It is known that CD4[+] Tfh cells play an important role in regulating B cell self-tolerance. Excessive CD4[+] Tfh cells can result in spontaneous GC generation and autoantibody production[19,20]. Stat5 negatively regulates CD4[+] Tfh cell differentiation and function, and Stat5 deficiency leads to increased CD4[+] Tfh cells, resulting in impaired B cell self-tolerance[29]. In our studies, the depletion of CD4[+] T cells completely abrogated GC B-cell formation following LCMV infection in wild-type but not CD8-specific Stat5-deficient mice. Consistently, CD4[+] T-cell-depleted CD8-specific Stat5-deficient mice displayed much higher levels of autoantibodies. However, in addition to CD4[+] Tfh cells, other cells, such as NKT, can help GC B-cell formation[39]. We found that NKT cells were selectively deleted in both Stat5-deficient and control mice following acute LCMV infection, consistent with the previous studies[41,42]. Our findings demonstrate that CD8[+] T-cell-specific deletion of Stat5 can enhance GC formation and specifically induce autoantibody production independent of CD4[+] T or NKT cells. It has been reported that CD8[+] T cells have potential B cell helper activities and are required for the formation of ectopic GCs in rheumatoid joint[36]. Here we identified CXCR5[+]PD-1[+]CD8[+] Tfh cells as a distinct novel CD8[+] T cell subset that can control B cell self-tolerance. These CXCR5[+]PD-1[+]CD8[+] Tfh cells specifically control autoreactive, but not antigen-specific, antibody production, suggesting that these CD8 T cells are not very potent at helping antigen-specific responses. However, the underlying mechanism of the regulatory specificity is not clear. Of note, a previous study has shown that CD8[+] regulatory T cells affect autoreactive, but not LCMV-specific, antibody production during

acute LCMV infection[40]. The development of CXCR5[+]PD-1[+]CD8[+] Tfh cells identified here was tightly controlled by the Stat5 pathway to maintain B cell self-tolerance. Dysregulation of CXCR5[+]PD-1[+]CD8[+] Tfh cell development by CD8[+] T-cell-specific deletion of Stat5 broke down B cell tolerance. The cytokine IL-21 is critical for CD4[+] Tfh cells to control B-cell function[25,64]. IL-21 contributes to development of autoimmune diseases[65]. IL-21 production is largely limited to activated CD4[+] T cells and NKT cells[66,67]. IL-21 is found to be expressed in Runx3-deficient CD8[+] Tfh-like cells[37], and to be required for the CD8[+] T cell to help to B cells[68]. Here we demonstrated that CXCR5[+]PD-1[+]CD8[+] Tfh cells were able to produce IL-21, which was enhanced by Stat5 deletion. Further investigation and characterization of CXCR5[+]PD-1[+]CD8[+] Tfh cells would help to develop new strategies to control autoimmunity by targeting follicular CD8 helper T cells.

## Methods

**Mice.** Mice with deletion of the entire Stat5A/5B locus (Stat5[−/−]) and mice with the entire Stat5A/5B locus gene flanked with loxP sites (Stat5[fl/fl]) were used[69]. The Stat5[fl/+] mice were bred with Stat5[+/−] mice to generate Stat5[fl/−] mice. CD8Cre transgenic mice, Ig[HEL] (C57BL/6, MD4), sHEL (C57BL/6, ML5) transgenic mice, BM1 mice, Rag1-deficient mice, CD45.1 congenic mice, CXCR5-deficient mice, PD-1-deficient mice, Lag3-deficieict mice and CD40-deficient mice were obtained from the Jackson Laboratory. Bcl6[fl/fl]CD4Cre mice were also used[70]. Mice were housed in the SPF animal facility at the Medical College of Wisconsin, the animal experiments were performed using protocols approved by the Institutional Animal Care and Use Committee.

**Acute LCMV infection.** Mice were infected with LCMV (Armstrong strain, $2 \times 10^5$ pfu/mouse) intraperitoneally.

**Flow cytometry and cell sorting.** Single-cell suspensions of thymus, spleen and BM cells were treated with Gey's solution to remove RBC and resuspended in PBS supplemented with 2% BSA. The cells were then stained with a combination of fluorescence-conjugated Abs. Apc-conjugated anti-B220 (17-0452) and anti-CD4 (17-0041), PE-Cy7-conjugated anti-B220 ((25-0452) and anti-CD8 (25-0081), PE-conjugated anti-CD40L (12-1541), and PerCP-efluor 710-conjugated anti-PD-1 (46-9985) were purchased from eBioscience. PE-conjugated anti-Fas (554258), Apc-conjugated anti-IgG1 (550874), Alexa Fluor 647-conjugated anti-GL7 (551529), and APC/CY7 conjugated anti-CD4 (552051) and anti-CD44 (103028) were purchased from BD Biosciences Pharmingen. APC-conjugated streptavidin (016-130-084) was purchased from Jackson ImmunoResearch Laboratories. APC-labeled CD1d Tetramer (mouse CD1d PBS-57) was kindly provided by NIH Tetramer Core Facility. The dilutions at which each antibody was used are included in Supplementary Table 3. Samples were applied to a flow cytometer (LSR II, Becton Dickinson) for FACS analysis or cell sorter (FACSAira IIIu, Becton Dickinson) for sorting. Data were collected and analyzed using FACSDiva software (Becton Dickinson). Flow cytometry gating strategies are shown in Supplementary Figs. 23 and 24.

**RNA-seq and transcriptome analysis.** The CD8[+] T-cell subsets were FACS sorted from the indicated mice and lysed in TRIzol (LifeTechnologies) to isolate total RNA. The mRNA was purified using NEBNext poly(A) mRNA Magnetic Isolate Module (New England Biolabs). The libraries were constructed using NEBNext Ultra RNA library Prep kit for Illumina (New England Biolabs), and quantified using QubitFluorometer (ThermoFisher) and Kapa Library Quantification Kit (KapaBiosystems). The average size of the libraries was measured using D1000 ScreenTape system (Agilent). A total of 1.7 pmol libraries were sequenced on Illumina NextSeq 500 with NextSeq 500/550 v2 kit. The raw reads of RNA-seq were preprocessed on the Illumina BaseSpace sequence hub and aligned to the mouse reference genome Mus musculus/mm10 (RefSeq) using the aligner STAR. Differentially expressed transcripts with FDR ≤ 0.05 were identified using the DESeq2 program as implemented on the Illumina BaseSpace hub. Heat maps were generated with normalized data of RNA-seq analysis. The GSEA was performed using GSEA software (http://software.broadinstitute.org). The gene sets for Tfh signature were created with genes that were differentially expressed and had a twofold higher expression in the Tfh cells relative to in Th1 cells (GEO accession code GSE67334)[63] or genes that were upregulated in Tfh-like CD8[+]Tim3[−]Blimp1[−] relative to CD8[+]Tim3[+]Blimp1[+] cells, TCF1-overexpressing relative to control cells, and TCF-1-deficient relative wild-type CD8[+] T cells (GEO accession code GSE85367)[51]. The gene sets for effector CD8 T cell signature were created with genes that were upregulated in CD8 effector T cells relative to memory CD8 T cells (EFF_VS_MEMORY_CD8_TCELL_UP, GSE1000002_1582_ 200_UP)[71].

**RT-qPCR analysis**. Splenic CXCR5⁻PD-1⁻CD8⁺, CXCR5⁺PD-1⁻CD8⁺, CXCR5⁻PD-1⁺CD8⁺ and CXCR5⁺PD-1⁺CD8⁺ T cells were sorted from Stat5⁺/⁺ CD8Cre and Stat5^fl/⁻CD8Cre mice 8 days after acute LCMV infection. Total RNA were isolated from the indicated cells and then cDNA was synthesized using Thermo Scientific Maxima First Strand cDNA Synthesis Kit (#K1642). RT-qPCR was performed using 7500 Real-Time system (Applied Biosystems). The following primer pairs were used: Il33 forward: 5′-TGAGACTCCGTTCTGGC CTC-3′, reverse: 5′-CTCT TCATGCTTGGTACCCGAT-3′; Il21 forward: 5′-TCAG CTCCACAAGATGTAAAGGG-3′, reverse: 5′-GGGCCACGAGGTCAATGAT-3′; Tcf7 forward: 5′-CTATCCCAGGTTCACCCACC-3′, reverse, 5′-TTCTCTGCCTT GGGTTCTGC-3′; Batf forward: 5′-GTTCTGTTTCTCCAGGTCC-3′, reverse: 5′-GAAGAATCGCATCGCTGC-3′; Bcl6 forward: 5′-GCCGGCTCAATAATCTC GTGAACAGGTCC-3′, reverse: 5′-CCAGCAGTATGGAGGCACATCTCTGT ATGC-3′; Eomes forward: 5′-GCCTACCAAAACACGGATA-3′, reverse: 5′-TCTG TTGGGGTGAGAGGAG-3′; and β-actin forward: 5′-CCACAGCTGAGAGGG AAATC-3′, reverse, 5′-CTTCTCCAGGGAGGAAGAGG-3′.

**IgG1 production in vitro**. B cells, CD4⁺ T cells and CD8⁺ T cell subsets were FACS sorted from LCMV-infected mice 8 days after infection. B cells ($1 \times 10^5$) were co-cultured with CD4⁺ or CD8⁺ T cells ($5 \times 10^4$) in 96-well plates in the presence of IL-4 (10 μg/ml). After 4 or 7 days in culture, IgG1⁺ cells were analyzed by FACS and the IgG1 level in the supernatants was measured by ELISA.

**scRNA-seq and analysis**. CXCR5⁺PD-1⁺CD8⁺ T cells were FACS sorted from the indicated mice into 10 μl of PBS with 0.04% of BSA and were loaded on the Chromium Controller (10x Genomics). A total of 1700 cells were subsequently applied to make DNA library using the Chromium Single Cell 3′ v2 reagent kit (10x Genomics) according to manufacturer's protocol. The libraries were quantified using QubitFluorometer (ThermoFisher) and Kapa Library Quantification Kit (KapaBiosystems). The average size of the libraries was measured using D1000 ScreenTape system (Agilent). A total of 1.7 pmol libraries were sequenced on Illumina NextSeq 500. Library preparation and sequencing for all samples were performed at the same time.

After sequencing, data for each sample were demultiplexed and aligned to the *Mus musculus* genome (mm10). UMI counts for each gene were quantified using Cell Ranger v1.3.1 (10x Genomics). Filtered barcode matrix files for each sample were then imported into the Seurat v2.2 single cell analysis package using R v3.4.3 (R Core Team, 2018)[72]. Cells expressing fewer than 200 genes and genes expressed in fewer than three cells were filtered out. Cells where mitochondrial transcripts comprised more than 7% of the cell's transcriptome were filtered out. After filtering, gene expression values for each cell were multiplied by a scale factor of 10,000 and log-normalized. A cell-cycle phase was predicted for each cell[73] and gene expression values were scaled based on the number of UMIs in each cell, the cell mitochondrial content, and the cell cycle score. Genes with high dispersion in at least two samples were used to conduct canonical correlation analysis to group the sub-populations of cells from each sample together (10.1101/164889). The first 20 canonical correlation vectors were chosen for downstream clustering and visualization based on their high shared correlation strengths and the genes driving each canonical correlation vector. The sample populations were aligned using these canonical correlation vectors and visualized using a t-distributed stochastic neighbor embedding plot (t-SNE). Naïve clustering of the cells into sub-populations was then conducted using Seurat's implementation of a shared nearest neighbor (SNN) modularity optimization-based clustering algorithm (Louvain's original algorithm described in 10.1140/epjb/e2013-40829-0). Differentially expressed genes between Stat5-sufficient and Stat5-deficient samples in specific clusters were determined using the MAST GLM framework (10.1186/s13059-015-0844-5). GSEA and GSVA were performed using the same gene sets for bulk RNA-seq data analysis, including GEO accession code GSE67334[63], GEO accession code GSE85367[51] and EFF_VS_MEMORY_CD8_TCELL_UP, GSE1000002_1582_200_UP[71].

**Adoptive transfer**. FACS-sorted B cells ($1.3 \times 10^6$ cells) from LCMV-infected BM1 mice were adoptively transferred into Rag1-deficient mice with CD8⁺ T cells ($9 \times 10^5$ cells) from acute LCMV-infected C57BL/6 or BM1 mice (day 8). The recipient mice were subsequently infected with LCMV (Armstrong, $2 \times 10^5$ pfu/mouse). Thirty days after infection, the IgG1 levels in the sera were measured by ELISA.

**Autoantibody profiling using autoantigen microarrays**. Autoantibody reactivates against a panel of 124 autoantigens were measured using an autoantigen microarray platform developed by University of Texas Southwestern Medical Center (https://microarray.swmed.edu/products/category/protein-array/). Briefly, serum samples were pretreated with DNAse-I and then diluted 1:50 in PBST buffer for autoantibody profiling. The autoantigen array bearing 124 autoantigens and four control proteins were printed in duplicates onto Nitrocellulose film slides (Grace Bio-Labs). The diluted serum samples were incubated with the autoantigen arrays, and autoantibodies were detected with Cy3-conjugated anti-mouse IgG (1:2000, Jackson ImmunoResearch Laboratories) and Cy5-conjugated anti-mouse IgM (1:2000, Jackson ImmunoResearch Laboratories), FITC-conjugated anti-mouse IgE (1:500, BioLegend) or Cy5-conjugated anti-mouse IgA (1:1000, Bioss Antibodies), using a Genepix 4200A scanner (Molecular Device) with laser wavelength of 532 nm and 635 nm. The resulting images were analyzed using Genepix Pro 7.0 software (Molecular Devices). The median of the signal intensity for each spot were calculated and subtracted the local background around the spot, and data obtained from duplicate spots were averaged. The background subtracted signal intensity of each antigen was normalized to the average intensity of the mouse IgG and IgM (or IgA and IgE), which were spotted on the array as internal controls. Finally, the net fluorescence intensity (NFI) for each antigen was calculated by subtracting a PBS control, which was included for each experiment as negative control. Signal-to-noise ratio (SNR) was used as a quantitative measurement of the true signal above background noise. SNR values ≥3 were considered significantly higher than background, and therefore true signals. The NFI of each autoantibody was used to generate heatmaps using Cluster and Treeview software (http://bonsai.hgc.jp/~mdehoon/software/cluster/software.htm). Each row in the heatmap represents an autoantibody and each column represents a sample. Yellow color represents the signal intensity higher than the mean value of the raw and green color means signal intensity is lower than the mean value of the raw. Gray or black color represents the signal is close or equal to the mean value of the raw.

**Statistical analysis**. Statistical analysis was performed with the unpaired two-tailed Student's *t*-test or the Mann–Whitney test.

**Reporting summary**. Further information on research design is available in the Nature Research Reporting Summary linked to this article.

## Data availability
The raw and processed total RNA and sc-RNA sequencing data files have been deposited in NCBI's Gene Expression Omnibus and are accessible through GEO Series accession number GSE136642. The source data underlying Figs. 1a, f, 2a–b, d, 3a–c, 4a, b, e, 7a–f and Supplementary Figs. 2a–c, 5, 6, 8b, c, and 13 are provided as a Source Data file.

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

## Acknowledgements

This work is supported in part by NIH grants AI079087 (D.W.), HL130724 (D.W.), AI125741 (W.C.) and AI132771 (A.L.D.).

## Author contributions

Y.C. contributed to research design, performed the research, analyzed the results and wrote the first draft of the manuscript. M.Y., Y.Z., G.F., G.X., W.Z. and L.L. performed some experiments. R.B. helped the RNA-seq and single-cell RNA-seq data analysis. Q.L. helped with the autoantibody screening experiment. A.D. provided critical Bcl-6-deficient mice. N.Z. helped the RNA-seq and single-cell RNA-seq experiments and data analysis. W.C. helped the LCMV infection and critically reviewed the manuscript. L.M. provided intellectual input. R.W. provided intellectual input, supervised the study and critically reviewed the manuscript. D.W. conceived and supervised the study, analyzed the results and wrote the manuscript.

## Competing interests

The authors declare no competing interests.
