## [Peer Review File · Nature Communications]

Reviewers' comments:

Reviewer #1 (Autoimmunity, Tfh)(Remarks to the Author):

Comments to the authors:

In this manuscript, the authors describe a population of CD8 T cells that express Tfh cell markers. Several other published papers have already described a population of CXCR5+CD8+ T cells that resemble Tfh cells in the ability to stimulate B cells, to promote Ag-specific antibody responses, and to generate and maintain follicles and GCs (Quigley MF. Eur J Immunol. 2007; Yang R. J Exp. Med 2016; Jiang H DNA Cell Biol 2017; Kang YM J. Exp. Med 2002; Kim HJ. PNAS 2011; Miles B. PLoS Pathog. 2016). While other papers have shown a function in controlling viral replication during chronic viral infection (He R. Nature 2016; Leong YA. Nat Immunology 2016; Im SJ Nature 2016). The main points that this manuscript is trying to make are that Stat5 negatively regulates a population of Tfh CD8 T cells and that this regulation is important to maintain B cell tolerance. Although the work is potentially interesting, several concerns arise from the experimental models used and the poor design of the experiments. More specifically:

1. Since mice in the experimental groups (stat5⁻/fICD8cre) contain non-CD8 T cells expressing half normal amounts of stat5, stat5⁻/+CD8cre should be used as controls in all the experiments. Also, authors need to compare the phenotype of stat5⁻/+ and stat5^{+/+} mice. The authors show that stat5⁻/fICD8cre produced more autoreactive IgG antibodies compared to stat5⁻/+CD8cre (Fig. 2c). How is the autoantibody production compared to stat5^{+/+} mice? Do the stat5^{+/+} mice have increased levels of autoantibodies than stat5^{+/+} mice? The authors never show the number, phenotype, and function of T regulatory cells, which are critically regulated by Stat5 signaling. Why the authors use stat5⁻/fICD8cre mice and not stat5^{fl}/fICD8cre mice? Do the stat5⁻/fICD8cre and stat5^{fl}/fICD8cre have the same phenotype?
2. The authors show that naive stat5⁻/fICD8cre have increased frequency of CXCR5+PD1+ cells in the CD8+ gate (Fig. 3a). First, the plots do not show a convincing population of CXCR5+PD1+ cells and the opinion of this reviewer is that the plots show background staining. Second, since the numbers of CD8 T cells are dramatically reduced in stat5⁻/fICD8cre mice, therefore the number of CXCR5+PD1+CD8 T cells are also drastically reduced in stat5⁻/fICD8cre compared to control mice. This is in contradiction of the overall hypothesis of the paper. The authors claim that stat5⁻/fICD8cre mice have increased CXCR5+PD1+CD8 T cells, resulting in the breakdown of B cell tolerance, however, the data show the opposite: stat5⁻/fICD8cre mice have reduced numbers of CXCR5+PD1+CD8 T cells. How is the frequency of CXCR5+PD1+CD8 T cells in 50% stat5⁻/+ and stat5^{+/+} chimeras?
3. In figure 3, the authors should show the effector CD8 T cell response to LCMV using tetramers, the viral load and the gating strategy for the analysis of GC B cells.
4. Figure 4 is not novel.
5. The RNAseq analysis does not show enhanced expression of BCL-6, TCF-3, TFC-7, STAT-3 in CXCR5+PD1+CD8 T cells vs. other populations of CD8 T cells as the authors claim. In addition, sorted CXCR5+PD1+ cells from stat5^{fl}/+ express high levels of CXCR5 than sorted CXCR5+PD1+ cells from stat5^{+/+}, suggesting that different population are being sorted.

Reviewer #2 (Cytokine signaling, Tfh, T biology)(Remarks to the Author):

In this manuscript, the authors describe a CXCR5+ CD8 T cell subset that functions like Tfh cells to help B cell germinal center responses and that is inhibited by STAT5 activity, similar to CXCR5+CD4+ Tfh cells. Starting with the observation that deletion of STAT5 in mature CD8 cells led to autoantibody production and anti-HEL antibodies in IgHELsHEL mice, the authors showed that CD8 cell loss of

STAT5 led to generation of a small population of CXCR5+PD1+CD8+ cells in IgHELsHEL mice. In response to acute LCMV infection, STAT5fl/-CD8Cre mice showed more CXCR5+PD1+CD8+ and GC B cells, (but no increases in CXCR5+PD1+CD4+ Tfh cells). Depletion of CD4 cells prevented GC formation in control mice in response to LCMV. However, some STAT5fl/-CD8Cre mice could develop (low percentages of) GCs and these mice had significantly more autoantibodies (but not anti-LCMV antibodies). In vitro, CXCR5+PD1+CD8+ (from both STAT5fl/-CD8Cre and STAT5+/+CD8Cre LCMV infected animals) helped B cells, increasing IgG1 production in cultures (but not to the extent that CXCR5+PD1+CD4+ Tfh cells did). RNAseq data of CXCR5+PD1+CD8+ cells from control LCMV-infected animals provided evidence that this population was distinct from other CD8 cells in infected mice and resembled that of Tfh cells. Evaluation of gene expression by gene set enrichment analyses of CXCR5+PD1+CD8+ from STAT5fl/-CD8Cre mice showed an even stronger similarity to Tfh cell gene expression. The authors then performed scRNA seq of this population from IgHELsHEL or LCMV-infected mice that either lacked or had STAT5 expression in CD8 cells, which confirmed their conclusions from RNAseq. Finally, the authors very elegantly show that help for B cells in vitro was dependent on CD40L expression. This is a nice manuscript that provides yet more evidence for the wide-range of lymphocyte responses, particularly in response to infection.

Comments:

The lack of induction of LCMV-specific antibodies in CD4-depleted mice suggests that CD8 cell populations are not very potent at helping antigen-specific responses. The authors should comment on this relative to the potential function of these cells.

Importantly, it should be noted that the connection to CXCR5+CD8+ cells in vivo is only correlative. Other cells can help B cells for GC induction, including NKT cells, which may not be depleted by anti-CD4 antibodies. Could the authors check for the presence of NKT help for GC from these mice or deplete NKT cells as well as CD4 cells in these mice? These caveats must be discussed.

Unfortunately, the least illuminating part of the paper is the scRNAseq data. First, it was not clear that the differences represent anything other than different percentages of contaminating populations, especially since Stat5+/+CD8Cre/YFPIgHELsHEL had very few cells in the analyses. It was also not clear why the HEL mice were included in this comparison. The authors should also state the purity of the cells sorted (if they have these data) and efforts to avoid batch effects which could affect their minor differences. Were the RNA and library preps and sequencing for all samples done at the same time? I assume so as that the samples are so similar, but this should be stated. Overall, it might have been more informative to compare scRNASeq on CD8+CXCR5+PD1+ cells derived from different experimental situations (chronic LCMV and other infections, see below), but that is beyond the scope of this paper. The authors should discuss the limitation of their analyses.

The authors cite multiple other papers describing CXCR5+CD8+ T cells, several in the context of chronic LCMV infection, and describe them all as distinct CXCR5+CD8+ T cell populations. It is not clear that these are all different populations. The authors should also cite a Science Immunology paper from Wu et al (2016) that describes a similar population in chronic LCMV infection, and also confirms by GSEA analyses that these TCF1+ CXCR5+ CD8 cells are transcriptionally very similar to Tfh cells. It would be informative to compare the gene expression signatures from these different CXCR5+CD8+ populations as that gene-expression patterns have been reported. While much of this is discussed in the discussion, comparison of the data sets would be useful and should be possible even cross-platforms.

Figure 6 would be more informative if comparisons of ChIP for active and repressive Histone marks from available data sets were included. STAT5 binding is seen at what would be predicted to be both

repressed and activated genes and these further analyses may be informative.

Minor issues:

The authors use a CD8-Cre line and state :“ CD8Cre-mediated Stat5 deletion occurred in about 30% of CD8+, but not in CD4-CD8- (DN), CD4+CD8+ (DP) or CD4+ thymocytes (Fig. 1b)”. It would be clearer if they stated CD8+CD4- (SP) and CD4+CD8- SP cells.

On the IgG autoantibodies, were isotypes evaluated?

The authors state that “Since we examined the effect of Stat5 deficiency on CD8+ T cells themselves, we used corresponding Stat5+/+ mice as controls. Indeed, a population of CXCR5+PD-1+CD8+ T cells was easily detected in Stat5fl/-CD8Cre/YFPiGHEL sHEL relative to Stat5+/+CD8Cre/YFPiGHELsHEL, mice (Fig. 3a).” There could be problems arising from other cells. Were STAT5+/- mice ever used as controls in these experiments? If not, this caveat should be mentioned.

Overall, this is an interesting and mostly well-performed manuscript that provides evidence for a STAT5-restriction of a CXCR5+PD1+CD8 cell population that can provide help for B cells in vitro and likely in vivo. My concerns are that: 1) the authors should discuss that other populations may contribute to their phenotypes in vivo and 2) the scRNAseq was unfortunately not very illuminating. It may also be possible to use publicly-available datasets to provide more insight into this population.

Reviewer #3 (Cytokine signaling, Tfh, STAT)(Remarks to the Author):

CD4 T cells that localize to B-cell follicles in secondary lymphoid tissues do so because they acquire expression of the B-cell zone homing chemokine receptor CXCR5. These cells are termed T follicular helper cells and they mediate the differentiation of B cells into memory and plasma cells, the effector cells of long-lived humoral immunity. Just like B cells and CD4 T cells, other cell types can also acquire expression CXCR5, namely NKT cells, gamma delta T cells, and Tregs (ie T follicular regulatory cells), which provides entry into follicles where they can then modulate the function of B cells.

Tfh cells have been implicated in the autoimmunity, as well as control of chronic viral infections in humans/primates, as well as murine models. Here, Tfh cells stimulate production of specific, or pathogenic autoreactive, Abs. It is therefore important to understand the balance between helper cells controlling tolerance as well as their contributions to protection against chronic infection.

Interestingly, some CD8 T cells have also been reported to express CXCR5 – the first report being by Quigely et al Eur J Immunol 2007 37: 3352. However, at this time (>10 yrs ago) the function and relevance of CD8 T cells acquiring expression of CXCR5 was unclear, especially in the context of infections and immune regulation. More recently, there has been a flurry of papers in Nature, Nature Immunol and elsewhere providing detailed analyses of CD8+ Tfh-type cells in the setting of murine models of chronic viral infection as well as in humans with similar chronic infections such as HIV or EBV. The requirements for the generation, maintenance and function of these cells has started to be elucidated from these recent studies, with CD8+ Tfh-like cells sharing many of the developmental and functional requirements as bona fide Tfh cells.

In this study, Chen et al report their findings that add to the growing body of data relating to CD8 Tfh-type cells. the novel findings presented here are that STAT5 appears to negatively regulate the

generation of CD8 Tfh-type cells inasmuch that in the absence of STAT5 there marked increases in the % of CD8 Tfh-type cells following LCMV infection. Interestingly, the levels of autoreactive IgG, but not IgM, were also increased following deletion of STAT5. In a model of B cell self tolerance, STAT5 deficiency caused a 2-3 fold increase in %'s of CD8 Tfh-like cells and GC B cells (tho the data for these readouts is highly variable). Whilst the levels of anti-LCMV Abs were not increased, the levels of autoAbs were heightened by STAT5 deficiency.

Both STAT5 sufficient and STAT5 deficient CD8 Tfh-type cells could provide help to B cells in vitro and this is likely due to the ability of these cells to acquire expression of CD40L, which has been reported to be detected in some CD8 T cells previously tho the function of it has been under-analysed. Transcriptomic analysis revealed many similarities between CD8 Tfh-like cells and genuine CD4 Tfh cells including expression of canonical Tfh genes as well as over and under expression of relevant regulatory genes. This is further revealed from scRNA analysis which revealed multiple subsets of CD8 Tfh-like cells.

Overall, there is a lot of information presented in this study that certainly adds to our understanding of the biology and function of CD8 Tfh-like cells, at least in mice. The study is exhaustive and detailed, but despite this several key findings contained within merely confirm previous studies on CD8 Tfh-like cells. However, this is still a valuable contribution to the field of Tfh and related cells underpinning B cell differentiation and disease such as autoimmunity.

Comments

1. this is a well performed study, however it suffers from many of the key findings basically confirming previous discoveries relating to CD8 Tfh cells. while the transcriptomics is clearly detailed and impressive and informative it can not be overlooked that similar approaches by other groups have arrived at similar conclusions relating to eg expression of canonical Tfh genes by CD8 Tfh-like cells, and likely requirements for these processes. If the authors are going to arrive at the same conclusions then they need to extend them by rigorously establishing the validity of their scRNA seq data in terms of either confirming differential expression of many of the genes identified (rather than just a few of the most obvious ones) or pursuing their predictions of the importance of some of the genes/pathways derived from the transcriptomic analyses for CD8 Tfh-like cell function ie knocking these unique genes out or down to test their relevance. This will greatly add to the novelty of this paper
2. the main and most exciting finding is the putative role of STAT5 in restraining CD8 Tfh-like cells and preventing the generation of autoreactive B cells and autoAbs. This is perhaps not surprising given parallel findings made previously for STAT5 deficiency and CD4 Tfh cells. But this is an area that could be pursued more to really add a key finding to this paper. It is appreciated that the authors used the MD4 HEL model which is not conducive with Ig class switching – however a striking finding is the detection of autoreactive IgG, but not IgM, in non-Tg mice. given the availability of other models of B cell self tolerance that are compatible with class switching it was surprising the authors did not look into these to confirm the pathogenic role of STAT5 deficient CD8 Tfh-type cells on the generation of class switched autoAbs in the mouse models.
3. There is quite a bit of variability in the results presented eg Fig 3A, B and D – there is clearly an overlap for some of the data points relating to the control and Stat5 deficient groups. The same holds for the RNA Seq data in Fig 5 eg Bcl6, Eomes and Tcf7 are elevated in the CXCR5+PD1+ CD8 T cells but this is really only the case for 1 of 3 tests. Same for IL21 and IL33. For these reasons these RNA Seq data really needs to be confirmed by qPCR on separate samples or ideally by FACS.
4. What is the significance of IL-33 – it is stated it has been implicated in IgA and autoimmunity but the authors clearly demonstrate that only IgG is autoreactive in the absence of STAT5. Was the specificity of IgA tested for self-Ags?

Reviewer #1:

Reviewer #1 mentions that a population of CD8 T cells has been reported to express Tfh cell markers, stimulate B cells, promote Ag-specific antibody responses, and generate and maintain follicles and GCs. Other studies have shown this sub-population of CD8 T cells can control viral replication during chronic viral infection. The reviewer thinks that the main points of our manuscript are that Stat5 negatively regulates a population of CD8 Tfh cells and that this regulation is important to maintain B cell tolerance. We appreciate the positive comments on the novelties of our findings and many constructive suggestions from Reviewer #1. We would like to emphasize that CXCR5⁺PD-1⁺CD8⁺ Tfh cells identified in our study are functionally and transcriptionally distinct from the previously identified CXCR5⁺CD8⁺ or CXCR5⁺PD-1⁺CD8⁺ T cells in mice with chronic LCMV infection (see the first paragraph in the discussion section of the revised manuscript). Specifically, the reviewer raises several concerns.

Responses to specific comments:

1. (1) *The reviewer points out that “mice in the experimental groups (stat5-^{fl}/CD8cre) contain non-CD8 T cells expressing half normal amounts of stat5.” Thus, “stat5-^{fl}/CD8cre should be used as controls in all the experiments.” (2) The reviewer also requests us to “compare the phenotype of stat5-^{fl}/+ and stat5^{+/+} mice.” We showed that “stat5-^{fl}/CD8cre produced more autoreactive IgG antibodies compared to stat5-^{fl}/+CD8cre (Fig. 2c).” Thus, the reviewer asks “how is the autoantibody production compared to stat5^{+/+}mice? Do the stat5^{+/+} mice have increased levels of autoantibodies than stat5^{+/+} mice?” (3) The reviewer also asks us to show “the number, phenotype, and function of T regulatory cells, which are critically regulated by Stat5 signaling.” (4) The reviewer questions us: “Why did we use stat5-^{fl}/CD8cre mice but not stat5^{fl}/CD8cre mice? Do the stat5-^{fl}/CD8cre and stat5^{fl}/CD8cre have the same phenotype?”*

(1) This is a reasonable concern that non-CD8 T cells express half normal amounts of Stat5 in the experimental mice (Stat5^{fl}/CD8cre). Therefore, we took tremendous effort to re-perform the experiments with Stat5^{+/-} mice as the controls. The new data confirm our previous findings using Stat5^{+/+} mice as the controls and verify our conclusion that Stat5 negatively regulates a subpopulation of CD8 Tfh cells. The new data have been added to **Figs. 1, 2, 3b & 3c** and **Supplementary Figs. 2 to 8**, and incorporated into the Results (pages 5-9) of the revised manuscript.

(2) We examined the lymphocyte development and autoantibody production in Stat5^{+/-} mice. We found that the development of CD8⁺ T cells, but not CD4⁺ T cells or B cells, was impaired in Stat5^{+/-} relative to Stat5^{+/+} mice. Nonetheless, the autoantibody production was comparable between Stat5^{+/+} and Stat5^{+/-} mice. These new data have been included in **Supplementary Figs. 2 and 4** and incorporated into the Results (page 6) of the revised manuscript.

(3) We examined the effect of Stat5 deletion in CD8 T cells on CD4 Treg cells. We found that CD8Cre-mediated Stat5 deletion in Stat5^{fl/-}CD8Cre/YFP mice specifically occurred in CD8⁺ T cells (**Figs. 1a, b, d**). The deletion of Stat5 in CD8⁺ T cells did not affect the development of CD4⁺ Treg cells. The new data have been added to **Supplementary Fig. 1** and incorporated into the Results (page 6) of the revised manuscript.

(4) The defects of CD8⁺ T cells in Stat5^{fl/-}CD8Cre mice were comparable to those in Stat5^{fl/fl}CD8Cre mice. The data are included in the **Figure** at the end of this letter for the reviewers. Therefore, we utilized Stat5^{fl/-}CD8Cre mice as our experimental mice.

2. The reviewer raises concerns about the observation “that naïve stat5-^{fl}CD8cre have increased frequency of CXCR5⁺PD1⁺ cells in the CD8+ gate (Fig. 3a).” (1) “The plots do not show a convincing population of CXCR5⁺PD1⁺ cells” and “the plots show background staining.” (2) Since the numbers of CD8 T cells are dramatically reduced in stat5-^{fl}CD8cre mice, therefore the number of CXCR5⁺PD1⁺ CD8 T cells are also drastically reduced in stat5-^{fl}CD8cre compared to control mice. This is in contradiction of the overall hypothesis of the paper. The authors claim that stat5-^{fl}CD8cre mice have increased CXCR5⁺PD1⁺ CD8 T cells, resulting in the breakdown of B cell tolerance, however, the data show the opposite: stat5-^{fl}CD8cre mice have reduced numbers of CXCR5⁺PD1⁺ CD8 T cells. (3) How is the frequency of CXCR5⁺PD1⁺ CD8 T cells in 50% stat5^{-/-} and stat5^{+/+} chimeras?

(1) CXCR5⁺PD-1⁺CD8⁺ T cells were a rare subpopulation. As shown in **Supplementary Fig. 21** of the revised manuscript, there were no CXCR5⁺PD-1⁺ populations in CXCR5-deficient or PD-1-deficient mice, demonstrating that the CXCR5⁺PD-1⁺ plots specifically detected a real cell population but were not background staining. In addition, CXCR5⁺PD-1⁺CD8⁺ T cells were a distinct subpopulation, especial in Stat5^{fl/-}CD8Cre mice, as shown in **Fig. 3c** of the revised manuscript.

(2) The numbers of CD8 T cells were markedly reduced in Stat5^{fl/-}CD8Cre relative to control mice. However, the percentages of CXCR5⁺PD-1⁺CD8⁺ within CD8⁺ population were dramatically increased. Consequently, the absolute numbers of CXCR5⁺PD-1⁺CD8⁺ T cells were comparable between Stat5^{fl/-}CD8Cre and control mice. These data are shown in **Fig. 3a** and described in the Results (page 7) of the revised manuscript. Further, after acute LCMV infection, the percentage and absolute number of CXCR5⁺PD-1⁺CD8⁺ T cells were markedly increased in Stat5^{fl/-}CD8Cre relative to control mice. These data are shown in **Fig. 3c** and described in the Results (page 8) of the revised manuscript. Furthermore, RNA-seq data demonstrated that Stat5-deficient relative to control CXCR5⁺PD-1⁺CD8⁺ T cells displayed increased expression of the Tfh-specific genes, indicating enhanced functions of the mutant CD8 Tfh cells (**Fig. 5** of the revised manuscript).

(3) Following acute LCMV infection, the frequency of CD45.2⁺CXCR5⁺PD-1⁺CD8⁺ T cells in BM chimeric Rag1-deficient mice that received the mixture of CD45.2⁺Stat5^{fl/-}CD8Cre mouse BM with wild-type CD45.1⁺ congenic mice BM was dramatically higher than that in control BM chimeric mice that received the mixture of CD45.2⁺Stat5^{+/-}CD8Cre mouse BM with wild-type CD45.1⁺ congenic mice BM. The new data have been added to **Supplementary Fig. 6** and incorporated into the Results (page 8) of the revised manuscript.

3. The reviewer requests us in Figure 3, to show the effector CD8 T cell response to LCMV using tetramers, the viral load and the gating strategy for the analysis of GC B cells.

As requested by the reviewer, we have included the frequency of antigen-specific CD8⁺ T cell detected by the MHCI-tetramer (GP33) staining, the viral clearance information, and the gating strategy of GC B cells in **Supplementary Fig. 5** and **Fig. 3b** and described in the Results (page 8) of the revised manuscript.

4. The reviewer thinks that Figure 4 is not novel.

We have removed Figure 4 to **Supplementary Fig. 8** of the revised manuscript.

5. The reviewer mentioned that “the RNAseq analysis does not show enhanced expression of BCL-6, TCF-3, TFC-7, STAT3 in CXCR5⁺PD1⁺CD8⁺ T cells vs other populations of CD8 T cells as the authors claim. In addition, sorted CXCR5⁺PD1⁺ cells from stat5fl/- express high levels of CXCR5 than sorted CXCR5⁺PD1⁺ cells from stat5+/, suggesting that different population are being sorted.”

The reason for highlighting Bcl-6, TCF-3, TCF-7 and Stat3 in original **Fig. 5a** was to indicate that these genes are related to CD4 Tfh cells. We did not claim that the expression of these genes was enhanced in CXCR5⁺PD1⁺CD8⁺ T cells relative to other populations of CD8 T cells. We apologize for the confusion. In **Fig. 5a** of the revised manuscript, we have removed highlights from these genes.

Stat5 suppresses the expression of many Tfh-specific genes, including CXCR5, in CXCR5⁺PD-1⁺CD8⁺ T cells. It is expected that Stat5-deficient and Stat5-sufficient CXCR5⁺PD-1⁺CD8⁺ T cells were different in terms of Tfh-related gene expression. Nonetheless, the Stat5-deficient and Stat5-sufficient CXCR5⁺PD-1⁺CD8⁺ T cells were the same type of cells, which was confirmed by high-throughput single-cell RNA-seq analysis. The t-SNE plot analysis of Stat5-deficient and Stat5-sufficient CXCR5⁺PD-1⁺CD8⁺ T cells demonstrated that the two populations were divided into same 5 clusters (**Fig. 6** of the revised manuscript), validating that both the populations were the same type of cells.

Reviewer #2:

Reviewer #2 thinks: “this is an interesting and mostly well-performed manuscript that provides evidence for a STAT5-restriction of a CXCR5⁺PD-1⁺CD8⁺ cell population that can provide help for B cells in vitro and likely in vivo.” We are grateful for the positive comments. The reviewer also gives several specific suggestions.

Responses to specific comments:

1. The reviewer points out: “the lack of induction of LCMV-specific antibodies in CD4-depleted mice suggests that the CD8 cell populations are not very potent at helping antigen-specific responses”. The reviewer suggests us to “comment on this relative to the potential function of these cells.”

We have provided comments in the Discussion section (page 20) of the revised manuscript.

2. The reviewer points out that “other cells can help B cells for GC induction, including NKT cells, which may not be depleted by anti-CD4 antibodies.” The reviewer asks us to check for the presence of NKT help for GC from these mice or deplete NKT cells as well as CD4 cells in these mice, and to discuss these caveats.

We have examined the number NKT cells in the spleen of Stat5^{fl/-}CD8cre and control mice after acute LCMV infection. Consistent with the previous findings that acute LCMV infection selectively leads to the loss of NKT cells through apoptosis (Journal of virology, 2001, 75: 10746; Eur J Immunol, 2005, 35;879-889), we found NKT cells were absent in both Stat5^{+/+}CD8Cre and Stat5^{fl/-}CD8cre mice after acute LCMV infection. Therefore, the increase of GC B cell formation and autoantibody production in CD4⁺ T cell-depleted Stat5^{fl/-}CD8cre mice was not due to the help from NKT cells. The new data have been added to **Supplementary Fig. 7c**, incorporated into the Results on page 9, and discussed on page 20 of the revised manuscript.

3. The reviewer provides several suggestions for presenting the scRNA-seq data. (1) The reviewer raises concern about the purity of the sorted cells. S/he thinks “it was not clear that the differences represent anything other than different percentages of contaminating populations, especially since Stat5^{+/+}CD8Cre/YFP⁺Ig^{HEL}sHEL had very few cells in the analyses,” and “it was also not clear why the HEL mice were included in this comparison.” (2) The reviewer asks us to take efforts to avoid batch effects which could affect their minor differences, and to state whether the RNA and library preps and sequencing for all samples were done at the same time. (3) S/he suggests that it might have been more informative to compare scRNASeq on CD8⁺CXCR5⁺PD1⁺ cells derived from different experimental situations (chronic LCMV and other infections), but s/he thinks that is beyond the scope of this paper. The reviewer suggest us to discuss the limitation of their analyses.

(1) This is a reasonable concern. We agree that CXCR5⁺PD-1⁺CD8⁺ T cells from Ig^{HEL}sHEL transgenic mice were very few. Now we have removed all of the scRNA-seq results of transgenic mice from original Fig. 6 to the **Supplemental Figs. 17-18** of the revised manuscript. Of note, the t-SNE plot analysis of the scRNA-seq data revealed that CXCR5⁺PD1⁺CD8⁺ T cells had a possible minor contamination of CD8⁺ dendritic cells that were in the cluster 5 and highly expressed Tyrobp, CD74 and other Class II genes. These results have been included in **Supplementary Figs. 14-18** and described in the Results (Pages 12-13) of the revised manuscript.

(2) The RNA isolation, library preparation and sequencing for all samples were done at the same time. This information has been included in the Methods section describing sc-RNA-seq and analysis on Page 24 of the revised manuscript.

(3) We agree that comparison of scRNA-Seq on CD8⁺CXCR5⁺PD-1⁺ cells derived from different experimental situations, such as chronic LCMV and other infections, is beyond the scope of this paper. As suggested, we have discussed the limitation of our analyses in the Discussion on pages 18-19 of the revised manuscript.

4. The reviewer points out that we “cite multiple other papers describing CXCR5+CD8+ T cells, several in the context of chronic LCMV infection, and describe them all as distinct CXCR5+CD8+ T cell populations” and suggests us to “cite a Science Immunology paper from Wu et al (2016) that describes a similar population in chronic LCMV infection.” In addition, the reviewer suggests us to “compare the gene expression signatures from these different CXCR5+CD8+ populations as that gene-expression patterns have been reported and thinks “the comparison of the data sets would be useful.”

As suggested, we have cited this Science Immunology article from Wu et al that has described the TCF1⁺Tim3⁺Blimp1⁻Cxcr5⁺ CD8 T cell population in chronic LCMV infection in the revised manuscript. We also performed comparative gene set enrichment analysis (GSEA) and gene set variation analysis (GSVA) of CD8⁺CXCR5⁺PD1⁺ with the gene signatures derived from the different Tfh-like CD8⁺ populations that have been reported. These new data have been added to **Figs. 6e, 6f** and **Supplementary Figs. 9, 19** and incorporated into the Results (pages 11-14) of the revised manuscript.

5. The reviewer advised that “Figure 6 would be more informative if comparisons of ChIP for active and repressive Histone marks from available data sets were included,” and that “STAT5 binding is seen at what would be predicted to be both repressed and activated genes and these further analyses may be informative.”

We thank the reviewer for this valuable advice. We compared the ChIP data for active histone marks, such as H3K4me1, H3K4me3, and H3K27ac, and repressive histone marks, such as H3K27me3, in CD8⁺ cells from acute LCMV-infected wild-type mice (Nat Immunol. 2017; 18: 931). We found the Stat5 binding regions within the Tfh-related genes largely overlapped with the peaks of histone marks. These new data have been added to **Supplementary Fig. 22** and incorporated into the Results (page 15) of the revised manuscript.

Minor issues:

1. The reviewer mentions that we use a CD8-Cre line and state: “CD8Cre-mediated Stat5 deletion occurred in about 30% of CD8+, but not in CD4-CD8- (DN), CD4+CD8+ (DP) or CD4+ thymocytes (Fig. 1b).” S/he thinks that “it would be clearer if they stated CD8+CD4- (SP) and CD4+CD8- SP cells.”

As suggested, we have changed CD8⁺ and CD4⁺ to CD8⁺CD4⁻ (CD8 SP) and CD4⁺CD8⁻ (CD4 SP) on Page 5 of the revised manuscript.

2. The reviewer asks whether we evaluated other autoantibody isotypes other than IgG.

In addition IgG autoantibodies, we measured IgA and IgE autoantibodies in Stat5^{fl/-}CD8^{cre} mice. The new data have been added to **Fig. 4c** and **Supplementary Fig. 3** and incorporated into the Results (pages 6 and 9) of the revised manuscript.

3. The authors state that “Since we examined the effect of Stat5 deficiency on CD8+ T cells themselves, we used corresponding Stat5^{+/+} mice as controls. Indeed, a population of CXCR5⁺PD-1⁺CD8⁺ T cells was easily detected in Stat5^{fl/-}CD8^{Cre}/YFP⁺Ig^{HEL}sHEL relative to Stat5^{+/+}CD8^{Cre}/YFP⁺Ig^{HEL}sHEL, mice (Fig. 3a).” There could be problems arising from other cells. Were STAT5^{+/-} mice ever used as controls in these experiments? If not, this caveat should be mentioned.

As we addressed Reviewer #1’s 1st question, we re-performed the experiments with Stat5^{+/-} mice as the controls. Due to the lack of enough numbers of Stat5^{+/-}Ig^{HEL}sHEL transgenic mice, Stat5^{+/+}Ig^{HEL}sHEL mice were still used as controls in Fig 3a. As suggested, the caveat has been mentioned on Page 17 of the revised manuscript. The new data have been added to **Figs. 1, 2, 3b & 3c** and **Supplementary Figs. 2 to 8**, and incorporated into the Results (pages 5-9) of the revised manuscript.

4. Overall, the reviewer thinks “this is an interesting and mostly well-performed manuscript that provides evidence for a STAT5-restriction of a CXCR5⁺PD1⁺CD8 cell population that can provide help for B cells in vitro and likely in vivo.” The reviewer’s main concerns are that: 1) the authors should discuss that other populations may contribute to their phenotypes in vivo and 2) the scRNAseq was unfortunately not very illuminating. It may also be possible to use publicly-available datasets to provide more insight into this population.

We are grateful for the very positive comments. Now, we 1) have discussed the potential roles of other cells, such as NKT cells, in Stat5^{fl/-}CD8^{cre} mice on page 20 of the revised manuscript and 2) have improved the scRNA-seq analysis and presentation. Moreover, GSEA analysis of our RNA-seq data with published datasets provided more insight into CXCR5⁺PD1⁺CD8⁺ T cells. These new data have been added to **Figs. 6e, 6f** and **Supplementary Figs. 9, 19** and incorporated into the Results (pages 11-14) of the revised manuscript.

Reviewer #3:

Reviewer #3 thinks: “there is a lot of information presented in this study that certainly adds to our understanding of the biology and function of CD8 Tfh-like cells, at least in mice.” “This is still a valuable contribution to the field of Tfh and related cells underpinning B cell differentiation and disease such as autoimmunity.” We appreciate the positive comments. The reviewer gives several specific suggestions.

Responses to specific comments:

1. The reviewer believes “this is a well performed study, however it suffers from many of the key findings basically confirming previous discoveries relating to CD8 Tfh cells.” “While the transcriptomics is clearly detailed and impressive and informative, it can not be overlooked that similar approaches by other groups have arrived at similar conclusions relating to eg expression

of canonical Tfh genes by CD8 Tfh-like cells, and likely requirements for these processes.” Therefore, the reviewer advises us to extend our study “by rigorously establishing the validity of their scRNA seq data in terms of either confirming differential expression of many of the genes identified or pursuing their predictions of the importance of some of the genes/pathways derived from the transcriptomic analyses for CD8 Tfh-like cell function ie knocking these unique genes out or down to test their relevance.” The reviewer believes that “this will greatly add to the novelty of this paper.”

We thank the reviewer for the important comment. We further used comparative GSEA and GSVA of scRNA-seq data to demonstrate that Tfh-specific genes were differentially expressed in cells in the cluster 1 relative to cluster 2 of CXCR5⁺PD-1⁺CD8⁺ T cells. The new data have been added to **Figs. 6e, 6f** and **Supplementary Figs. 19** and incorporated into the Results (pages 13-14) of the revised manuscript. We used real-time quantitative PCR to verify that the expression of Il-21, Il-23, Bcl6, Tcf7, Batf and Eomes was consistent with scRNA-seq data. The new data have been added to **Supplementary Fig. 13** and incorporated into the Results (page 12) of the revised manuscript. We also used FACS analysis to confirm that the expression of Bcl6, Tcf7, Batf, Icos, Tim3 and Lag3 was increased in Stat5-deficient relative to wild-type CXCR5⁺PD-1⁺CD8⁺ T cells. The new data have been added to **Supplementary Figs. 10 and 11** and incorporated into the Results (page 12) of the revised manuscript.

Furthermore, we used several gene knock-out mice to determine the importance of the differentially expressed genes in Stat5-deficient CXCR5⁺PD-1⁺CD8⁺ T cells. We found that CXCR5, PD-1, Bcl6 or Lag3 deficiency impaired the generation of CXCR5⁺PD-1⁺CD8⁺ T cells following acute LCMV infection. The new data were included in the Results section (page 14 and **Supplementary Fig. 21**) and discussed (page 19) in the revised manuscript.

2. The reviewer points out that “the main and most exciting finding is the putative role of STAT5 in restraining CD8 Tfh-like cells and preventing the generation of autoreactive B cells and autoAbs. This is perhaps not surprising given parallel findings made previously for STAT5 deficiency and CD4 Tfh cells.” The reviewer advises us to pursue more to really add a key finding to this paper. S/he thinks our “striking finding is the detection of autoreactive IgG, but not IgM, in non-Tg mice.” S/he asks us to “look into these to confirm the pathogenic role of STAT5-deficient CD8 Tfh-type cells on the generation of class switched autoAbs in the mouse models.”

We thank the reviewer for the advice. We further examined the serum levels of IgA and IgE autoantibodies. We found that the levels IgE but not IgA autoantibodies were markedly increased in Stat5^{fl/-}CD8Cre/YFP mice relative to Stat5^{+/-}CD8Cre/YFP mice. The new data have been added to **Fig. 4c** and **Supplementary Fig. 3** and incorporated into the Results (pages 6 and 9) of the revised manuscript.

3. There is quite a bit of variability in the results presented eg Fig 3A, B and D – there is clearly an overlap for some of the data points relating to the control and Stat5 deficient groups. The same holds for the RNA Seq data in Fig 5 eg Bcl6, Eomes and Tcf7 are elevated in the CXCR5+PD1+ CD8 T cells but this is really only the case for 1 of 3 tests. Same for IL21 and IL33. For these reasons these RNA Seq data really needs to be confirmed by qPCR on separate samples or ideally by FACS.

We used real-time quantitative PCR to verify that the expression of Il-21, Il-23, Bcl6, Tcf7, Batf and Eomes was consistent with scRNA-seq data. The new data have been added to **Supplementary Fig. 13** and incorporated into the Results (page 12) of the revised manuscript. We also used FACS analysis to confirm that the expression of Bcl6, Tcf7, Batf, Icos, Tim3 and Lag3 was increased in Stat5-deficient relative to wild-type CXCR5⁺PD-1⁺CD8⁺ T cells. The new data have been added to **Supplementary Figs. 10 and 11** and incorporated into the Results (page 12) of the revised manuscript.

4. What is the significance of IL-33 – it is stated it has been implicated in IgA and autoimmunity but the authors clearly demonstrate that only IgG is autoreactive in the absence of STAT5. Was the specificity of IgA tested for self-Ags?

IL-33 is implicated in the production of IgA and IgE antibodies. We examined the serum levels of IgA and IgE autoantibodies. We found that the levels IgE but not IgA autoantibodies were markedly increased in Stat5^{fl/fl}-CD8Cre/YFP mice relative to Stat5^{+/+}-CD8Cre/YFP mice. The new data have been added to **Fig. 4c** and **Supplementary Fig. 3** and incorporated into the Results (pages 6 and 9) of the revised manuscript.

Figure for the reviewers. Stat5^{fl/fl}CD8Cre and Stat5^{fl/-}CD8Cre mice have comparable defects. Splenocytes from Stat5^{+/+}CD8Cre, Stat5^{fl/fl}CD8Cre and Stat5^{fl/-}CD8Cre mice were stained with anti-CD4, anti-CD8, and anti-B220 antibodies. Numbers indicate percentages of YFP⁺CD8⁺ (a), CD4⁺ and B220⁺ (b) cells in the live splenocytes (left) and each dot represents an individual mouse (right).

Reviewers' comments:

Reviewer #1 (Remarks to the Author):

Although the authors have performed additional experiments to support their main conclusion: "a distinct novel population of CD8 T cells expressing PD1 and CXCR5 controls auto-reactive germinal center formation and autoantibody production"; the experiments are still not conclusive.

In the manuscript, the authors show that naïve and infected *stat5fl/flCD8cre* mice have increased autoantibody production compared with the controls. The authors claim that this is due to an increasing presence of follicular CXCR5+PD1+CD8 T cells. However, this has never been directly demonstrated. Is the effect observed in vivo in germinal centers and autoantibody production dependent on CD8 T cells?, dependent on CXCR5+PD1+CD8 T cells? The authors should perform depletion and adoptive transfer experiments. Do the follicular CXCR5+PD1+CD8 T cells interact with GC B cells or do they stimulate extrafollicular antibody responses?

This reviewer still believes that the data do not support the conclusion that excessive accumulation of follicular CXCR5+PD1+CD8 T cells breaks B cell tolerance since naïve *stat5⁻/flCD8cre* and control mice have the same number of follicular CXCR5+PD1+CD8 T cells.

The author did not provide explanation and experiments to support the use of *stat5⁻/flCD8cre* mice and not *stat5fl/flCD8cre* mice. Do the *stat5⁻/flCD8cre* and *stat5fl/flCD8cre* have the same phenotype?

In supplementary figure 12, the authors show that *stat5⁻/flCD8cre* mice have impaired effector CD8 T cell responses (reduced IFN γ production) after LCMV infection. How this affects the viral titres in several organs and the overall B cell response?

Reviewer #2 (Remarks to the Author):

This manuscript is much improved with additional controls and experiments. While the scRNAseq still suffers from limitations where major differences may be due to contamination, (it might have been better to examine a larger population such as CD8s or CXCR5+CD8 cells), overall the paper is much improved.

Reviewer #3 (Remarks to the Author):

The authors have gone to considerable lengths to address the concerns raised during the initial review. The efforts put into revising and improving this manuscript are recognised and appreciated by this reviewer. A few questions/comments remain.

1. To determine the relevance of differential gene expression determined for CD8+ Tfh type cells in the presence/absence of Stat5, the authors tested the effects of deletion of some of these genes on the generation of such cells. The authors report that

"We examined the effect of CXCR5, PD-1, Bcl6 or Lag3 deficiency on CXCR5+PD-1+CD8+ T cell generation. CXCR5 or PD-1 deficiency resulted in no detection of CXCR5+PD-1+CD8+ T cell following acute LCMV infection (Supplementary Fig. 21)."

As the CD8+ Tfh type cells are defined as “CXCR5+PD1+CD8+ T cells”, it is not particularly surprising that deficiency of CXCR5 or PD1 resulted in the loss of these cells. Were alternative markers, identified in this study, used to indirectly detect these cells in the absence of CXCR5 or PD1 due to gene knock out?

2. Outcomes of statistical analyses of the data appears to be missing from some figures eg Fig 4A, Fig 7A-E. these should be included.

3. given the substantial interest in the area of CD8+ Tfh-type cells, and the emerging controversies/differences arising in publications by various groups that have identified and studied these cells – ie as outlined here by the authors, there seems to be several functionally and molecularly distinct populations or subsets of CD8 Tfh cells – it would be valuable to include a summary table comparing and contrasting these different subsets so give the reader a clear(er) view of where the cells described here by the authors sit in the landscape of “CD8+CXCR5+PD1+ cells” relative to those reported over the past few years by others.

Responses to specific comments:

Reviewer #1:

Reviewer #1 thinks that “although the authors have performed additional experiments to support their main conclusion: ‘a distinct novel population of CD8 T cells expressing PD1 and CXCR5 controls auto-reactive germinal center formation and autoantibody production’, the experiments are still not conclusive.” In the manuscript, the authors show that naïve and infected $Stat5^{fl/fl}CD8^{cre}$ mice have increased autoantibody production compared with the controls. The authors claim that this is due to an increasing presence of follicular $CXCR5^+PD1^+$ CD8 T cells. However, this has never been directly demonstrated. The reviewer is wondering: is the effect observed in vivo in germinal centers and autoantibody production dependent on CD8 T cells? dependent on $CXCR5^+PD1^+$ CD8 T cells?

Thus, the reviewer (1) requests us to “perform depletion and adoptive transfer experiments” and (2) asks whether “the follicular $CXCR5^+PD1^+$ CD8 T cells interact with GC B cells or they stimulate extrafollicular antibody responses”.

We appreciate the additional constructive suggestions from this reviewer. We found that $CXCR5^+PD1^+$, rather than $CXCR5^-PD1^-$, $CXCR5^+PD1^-$ or $CXCR5^-PD1^+$, CD8 T cells could directly stimulate B cells to produce antibodies in vitro (Supplementary Fig. 8). We performed the adoptive transfer experiments and found CD8 T cells could assist B cells to produce antibodies in vivo (Fig. 7f). In addition, we found that after depletion of CD4 T cells, $Stat5^{fl/-}CD8^{Cre}$, relative to control, mice could generate more germinal center B cells and produced more autoantibodies during LCMV infection (Fig. 4). To further address the reviewer’s concern, we measured the autoantibody production in Rag1-deficient recipients following CD8 T cell and B cell adoptive transfer and found that CD8 T cells helped B cells to generate autoantibodies. These new data have been added to Fig. 4e and incorporated into the Results (page 10) of the revised manuscript.

Moreover, we performed the in situ immunofluorescence staining experiments and found that CD8 T cells were localized in the B cell zones, especially in the GCs of LCMV-infected $Stat5^{fl/-}CD8^{Cre}$ and control mice. Consistent with the increased $CXCR5^+PD1^+$ CD8 T cells in $Stat5^{fl/-}CD8^{Cre}$ mice, markedly more CD8 T cells were found in the GCs of LCMV-infected $Stat5^{fl/-}CD8^{Cre}$ relative to control mice. These findings indicate that $CXCR5^+PD1^+$ CD8 T cells could interact with GC B cells. These new data have been added to Fig. 4d and incorporated into the Results (page 10) of the revised manuscript.

This reviewer still believes that the data do not support the conclusion that excessive accumulation of follicular CXCR5⁺PD1⁺ CD8 T cells breaks B cell tolerance since naïve stat5⁻/flCD8cre and control mice have the same number of follicular CXCR5⁺PD1⁺ CD8 T cells.

The absolute numbers of CXCR5⁺PD-1⁺CD8⁺ T cells were comparable between Stat5^{fl/-}CD8Cre and control mice (Fig. 3a). Importantly, RNA-seq data demonstrated that Stat5-deficient relative to control CXCR5⁺PD-1⁺CD8⁺ T cells displayed increased expression of the Tfh-specific genes, indicating enhanced functions of the mutant CD8 Tfh cells (Fig. 5). Further, after acute LCMV infection, both the percentage and absolute number of CXCR5⁺PD-1⁺CD8⁺ T cells were markedly increased in Stat5^{fl/-}CD8Cre relative to control mice (Fig. 3c). These data support the conclusion that excessive accumulation and/or enhanced function of follicular CXCR5⁺PD1⁺ CD8 T cells breaks B cell tolerance.

The author did not provide explanation and experiments to support the use of stat5⁻/flCD8cre mice and not stat5^{fl/fl}/flCD8cre mice. Do the stat5⁻/flCD8cre and stat5^{fl/fl}/flCD8cre have the same phenotype?

Stat5^{fl/-}CD8Cre and Stat5^{fl/fl}CD8Cre mice had the same CD8 T cell phenotype. Thus, we utilized Stat5^{fl/-}CD8Cre mice as our experimental mice. These data have been added to Supplementary Fig. 2b,c and incorporated into the Results (page 6) of the revised manuscript.

In supplementary figure 12, the authors show that stat5⁻/flCD8cre mice have impaired effector CD8 T cell responses (reduced IFN γ production) after LCMV infection. How this affects the viral titles in several organs and the overall B cell response?

We found that naïve Stat5^{fl/-}CD8Cre mice had similar basal levels of Igs compared to Stat5^{+/-}CD8Cre mice (Fig. 2b). After CD4 T cell depletion, Stat5^{fl/-}CD8Cre mice displayed the same levels of anti-LCMV specific antibodies compared to control mice (Fig. 4b). These findings demonstrate that the reduction of IFN γ production Stat5^{fl/-}CD8Cre mice following LCMV infection does not affect the overall B cell response. These results are described in the Results (page 9) of the revised manuscript.

We also measured the viral titers in the spleen, liver and kidney of Stat5^{fl/-}CD8Cre and Stat5^{+/-}CD8Cre mice after LCMV infection. We found that the viruses were cleared in all the organs in both Stat5^{fl/-}CD8Cre and control mice at day 8 after infection. Thus, reduced IFN γ production Stat5^{fl/-}CD8Cre mice does not affect viral clearance. These results are described in the Results (page 8) of the revised manuscript.

Reviewer #2:

Reviewer #2 thinks our “manuscript is much improved with additional controls and experiments. While the scRNA-seq still suffers from limitations where major differences may be due to

contamination, (it might have been better to examine a larger population such as CD8s or CXCR5+CD8 cells), overall the paper is much improved.”

The t-SNE plot analysis of the scRNA-seq data revealed that CXCR5⁺PD1⁺CD8⁺ T cells contain two main distinct subpopulations, the major fraction of Cxcr5^{hi}Tcf7^{hi}Bcl6^{hi}Btla^{hi}Icos⁺Maf⁺Pdcd1⁺prdm1^{lo} Tfh-like cells (cluster 1, ~80%) and a minor fraction of Cxcr5^{lo}gzma⁺gzmb⁺ effector-like cells (cluster 2, ~15%) (Fig. 6d and Supplementary Fig. 14-18). The clusters 3 to 5 accounted for very few cells (Fig. 6a-c). Nonetheless, the possible contamination of a few minor cell populations would not affect the conclusion that CXCR5⁺PD-1⁺CD8⁺ T cells are largely a novel T helper cell sublineage and are negatively regulated by Stat5.

Reviewer #3:

Reviewer #3 points out that we have gone to considerable lengths to address the concerns raised during the initial review. The efforts put into revising and improving this manuscript are recognized and appreciated by this reviewer. A few questions/comments remain.

1. To determine the relevance of differential gene expression determined for CD8+ Tfh type cells in the presence/absence of Stat5, the authors tested the effects of deletion of some of these genes on the generation of such cells. The authors report that “we examined the effect of CXCR5, PD-1, Bcl6 or Lag3 deficiency on CXCR5+PD-1+CD8+ T cell generation. CXCR5 or PD-1 deficiency resulted in no detection of CXCR5+PD-1+CD8+ T cell following acute LCMV infection (Supplementary Fig. 21).” As the CD8+ Tfh type cells are defined as “CXCR5+PD1+CD8+ T cells”, it is not particularly surprising that deficiency of CXCR5 or PD1 resulted in the loss of these cells. Were alternative markers, identified in this study, used to indirectly detect these cells in the absence of CXCR5 or PD1 due to gene knock out?

CXCR5- and PD1-deficient T cells were used as staining controls to demonstrate that CXCR5⁺PD1⁺CD8⁺ T cells we identified were a real CD8 subpopulation. We have added this description to the results of the revised manuscript on page15.

As suggested by the reviewer, we stained splenocytes from LCMV-infected PD1-deficient mice with other Tfh markers, such as ICOS. We found that PD1 deficiency did not affect the generation of CXCR5⁺ICOS⁺ CD8 T cells. Thus, PD1 seems not to be required for the generation of CD8 Tfh cells. CXCR5 is critical for the migration of Tfh cells into the follicular B cell zones. Further studies will be required to confirm the role of CXCR5 in the generation of CD8 Tfh cells. The new data have been added into Supplementary Fig. 21 and described in the Results of the revised manuscript on page15.

2. Outcomes of statistical analyses of the data appears to be missing from some figures eg Fig 4A, Fig 7A-E. these should be included.

The p values have been included in these figures in the revised manuscript.

3. Given the substantial interest in the area of CD8+ Tfh-type cells, and the emerging

controversies/differences arising in publications by various groups that have identified and studied these cells – ie as outlined here by the authors, there seems to be several functionally and molecularly distinct populations or subsets of CD8 Tfh cells – it would be valuable to include a summary table comparing and contrasting these different subsets so give the reader a clear(er) view of where the cells described here by the authors sit in the landscape of “CD8+CXCR5+PD1+ cells” relative to those reported over the past few years by others.

A summary table (Supplementary Table 2) has been included in the Discussion (page 18) of the revised manuscript.

REVIEWERS' COMMENTS:

Reviewer #1 (Remarks to the Author):

Although the authors have done one additional experiment to answer my previous comments, the results are still inconclusive. To demonstrate that the autoimmune phenotype of stat5fl/flCD8cre mice is due to an increased presence of follicular CXCR5+PD1+CD8 T cells, depletion and adoptive transfer experiments of the different antigen-specific/activated populations of CD8 T cells should be performed to analyze the resulting germinal center/ B cell response.

How and Why naïve stat5fl/flCD8cre mice have increased autoantibody production, despite the fact that they have the same number of follicular CXCR5+PD1+CD8 T cells than control mice?

The authors use the incorrect control mice (i.e., Stat5+/+CD8Cre instead of Stat5+/-CD8Cre) in the Experiments in Figures 3,4,5 and 6.

The authors do not show viral titres in several organs.

Our specific responses to the remaining concerns from reviewer #1 are as follows:

1. *“Although the authors have done one additional experiment to answer my previous comments, the results are still inconclusive. To demonstrate that the autoimmune phenotype of *stat5^{fl/fl}CD8^{cre}* mice is due to an increased presence of follicular CXCR5⁺PD1⁺CD8 T cells, depletion and adoptive transfer experiments of the different antigen-specific/activated populations of CD8 T cells should be performed to analyze the resulting germinal center/ B cell response.”*

This concern is the same additional concern raised by this reviewer after reviewing 1st revision of our manuscript. We have addressed this concern in our 2nd revision of our manuscript. We have addressed this concern in our 2nd rebuttal letter as follows:

We found that CXCR5⁺PD1⁺, rather than CXCR5⁺PD1⁻, CXCR5⁺PD1⁻ or CXCR5⁻PD1⁺, CD8 T cells could directly stimulate B cells to produce antibodies in vitro (**Supplementary Fig. 8**). We performed the adoptive transfer experiments and found CD8 T cells could assist B cells to produce antibodies in vivo (**Fig. 7f**). In addition, we found that after depletion of CD4 T cells, Stat5^{fl/-}CD8Cre, relative to control, mice could generate more germinal center B cells and produced more autoantibodies during LCMV infection (**Fig. 4**). To further address the reviewer's concern, we measured the autoantibody production in Rag1-deficient recipients following CD8 T cell and B cell adoptive transfer and found that CD8 T cells helped B cells to generate autoantibodies. These new data have been added to **Fig. 4e** and incorporated into the Results (page 10) of the revised manuscript.

Moreover, we performed the in situ immunofluorescence staining experiments and found that CD8 T cells were localized in the B cell zones, especially in the GCs of LCMV-infected Stat5^{fl/-}CD8Cre and control mice. Consistent with the increased CXCR5⁺PD1⁺ CD8 T cells in Stat5^{fl/-}CD8Cre mice, markedly more CD8 T cells were found in the GCs of LCMV-infected Stat5^{fl/-}CD8Cre relative to control mice. These findings indicate that CXCR5⁺PD1⁺ CD8 T cells could interact with GC B cells. These new data have been added to **Fig. 4d** and incorporated into the Results (page 10) of the revised manuscript.

2. *“How and Why naïve *stat5^{fl/fl}CD8^{cre}* mice have increased autoantibody production, despite the fact that they have the same number of follicular CXCR5⁺PD1⁺CD8 T cells than control mice?”*

This concern is also the same concern raised by this reviewer after reviewing 1st revision of our manuscript. We have addressed this concern in our 2nd revision of our manuscript. We have addressed this concern in our 2nd rebuttal letter as follows:

The absolute numbers of CXCR5⁺PD-1⁺CD8⁺ T cells were comparable between Stat5^{fl/-}CD8Cre and control mice (**Fig. 3a**). Importantly, RNA-seq data demonstrated that Stat5-deficient relative to control CXCR5⁺PD-1⁺CD8⁺ T cells displayed increased expression of the Tfh-specific genes, indicating enhanced functions of the mutant CD8 Tfh cells (**Fig. 5**). Further, after acute LCMV infection, both the percentage and absolute number of CXCR5⁺PD-1⁺CD8⁺ T cells were markedly increased in Stat5^{fl/-}CD8Cre relative to control mice (**Fig. 3c**). These data

support the conclusion that excessive accumulation and/or *enhanced function* of follicular CXCR5⁺PD1⁺ CD8 T cells breaks B cell tolerance.

3. *The authors use the incorrect control mice (i.e., Stat5^{+/+}CD8Cre instead of Stat5^{+/-}CD8Cre) in the Experiments in Figures 3,4,5 and 6.*

This concern is similar to #1 concern, “Since mice in the experimental groups (stat5⁻/flCD8cre) contain non-CD8 T cells expressing half normal amounts of stat5, stat5⁻/+CD8cre should be used as controls in all the experiments”, raised by this reviewer after reviewing our original manuscript. We have addressed this concern in our 1st revision of our manuscript and our 1st rebuttal letter. The reviewer was satisfied with our specific responses in the 1st revision of our manuscript because this reviewer did not raise any concern about this issue in his 2nd round of comments/critics.

During our 1st revision of our manuscript, we took tremendous effort to re-perform almost all of the experiments with Stat5^{+/-} mice as the controls. The new data confirm our previous findings using Stat5^{+/+} mice as the controls and verify our conclusion that Stat5 negatively regulates a subpopulation of CD8 Tfh cells. The new data with Stat5^{+/-} mice as the controls have been added to **Figs. 1, 2, 3b & 3c** of the revised manuscript. Although **Fig 4** has data with Stat5^{+/+} mice as the controls, the Stat5^{+/-} control data have been added to **Supplementary Figs. 3 & 4**. Additional data with Stat5^{+/-} as controls have been added to **Supplementary Figs. 2 & 5 to 8**. These new data with Stat5^{+/-} mice as the controls support our conclusion that Stat5 intrinsically regulates a subpopulation of CD8 Tfh cells. Moreover, **Supplementary Fig. 2** shows that Stat5^{+/-}, relative Stat5^{+/+}, mice had normal populations of CD4⁺ T or B cells. **Supplementary Fig. 4** reveals that the levels of autoantibodies were comparable between Stat5^{+/-} and Stat5^{+/+} mice. **Supplementary Fig. 6** demonstrates that in the presence of wild-type cells, Stat5-deficient CXCR5⁺PD-1⁺CD8⁺ T cells still markedly increased in BM chimeric mice following acute LCMV infection. These data further strongly support our conclusion that Stat5 intrinsically regulates CD8 Tfh cells. With substantial strong evidences supporting our main points of this manuscript, we think that RNA-seq analysis (**Fig 5**) and single-cell RNA-seq analysis (**Fig 6**) of Stat5^{+/-} but not Stat5^{+/+} CXCR5⁺PD-1⁺CD8⁺ T cells do not affect our findings of Tfh-like gene expression profile in CXCR5⁺PD-1⁺CD8⁺ T cells and enhanced Tfh-like gene expression by Stat5 deficiency in CXCR5⁺PD-1⁺CD8⁺ T cells. In summary, we have spent tremendous effort to address this concern in the 1st revision of our manuscript.

4. *“The authors do not show viral titles in several organs.”*

This concern is the same as the last additional raised by this reviewer after reviewing 1st revision of our manuscript. We have addressed this concern in our 2nd revision of our manuscript. We have addressed this concern in our 2nd rebuttal letter as follows:

We also measured the viral titers in the spleen, liver and kidney of Stat5^{fl/-}CD8Cre and Stat5^{+/-}CD8Cre mice after LCMV infection. We found that the viruses were cleared in all the organs in both Stat5^{fl/-}CD8Cre and control mice at day 8 after infection. Thus, reduced IFN γ

production Stat5^{fl/-}CD8Cre mice does not affect viral clearance. These results are described in the Results (page 8) of the revised manuscript.